# Sectoral Transformation of the Economic System during Crisis and Stable Growth Periods (A Case Study of the European Countries)

**Sergey Mikhailovich Vasin** 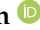

Department "Economic Theory and International Relations", Penza State University, 440026 Penza, Russia; pspu-met@mail.ru

**Abstract:** Sectoral structures are one of the critical and dynamic components of any social system subjected to either external or internal factors. The purpose of the paper is to reveal and validate characteristic features of transformation regarding economic sectoral structures during the crisis of the socio-economic system and the period of its coordinated development in order to determine the most stable industry sectors. This paper reveals the specificity of sectoral transformations in European countries during the crisis of 2008–2009 and the stable growth of 2010–2019. The analysis is premised on Robert B. Reich's sectoral structure, comprising production, in-person, intellectual, and communication services sectors. To conduct the research, statistical data analysis using the Gatev coefficient, and correlation and comparative analysis are applied. It is concluded that the mutable nature of sectoral dynamics depends on the planned changes resulting either from business expectations and interests, or state intervention. Yet, transformation is likely due to external and internal shocks (economic upheavals and wars), and unexpected events (natural disasters, epidemics, and pandemics). Over the last 15 years, the sectoral structure has been subjected to most of the above collisions. In-person, intellectual, and communication services sectors are least affected during the economic crisis. In the European countries, the period of economic growth is characterized by the growing dominance of intellectual and communication services sectors. There is a trend of decreasing the share of mining and quarrying in the sectoral production structure in favor of manufacturing industries and services.

**Keywords:** sectoral structure; sectoral dynamics; transformation; economic growth; economic crisis

## 1. Introduction

The sectoral structure is considered to be one of the decisive factors in achieving high productivity of the state. According to Collin M. Constantine, "it is not institutions that cause growth; rather, it is a country's economic structure that is the fundamental cause of economic performance. Therefore, differences in economic structures across time and space can explain the differences in economic development" (Constantine 2017).

To determine the adjusting elements for the dynamics models of the sectoral structure, it is necessary to:

- Introduce a concept of the sectoral structure;
- Construct a theoretical model for the sectoral structure;
- Present the dynamics of changes in the sectoral structure in the context of systematic development;
- Display changes in the sectoral structure in the context of major transformations;
- Determine transformation vectors for sectoral dynamics models.

Provided these problems are solved, it is feasible to advance in building an efficient and stable sectoral structure resistant both to predictable crisis phenomena and to sudden impacts affecting the socio-economic system.

The purpose of the paper is to reveal and validate characteristic features of transformation regarding the economic sectoral structure during the crisis of the socio-economic system and the period of its systematic development in order to determine the most stable industry sectors.

The specification of regular transformations of the sectoral structure and identification of the most stable industry sectors under varying factors of economic dynamics (crisis and stable growth periods) are considered as the added value of the paper.

## 2. Literature Review

It is generally accepted that the sectoral structure is the ratio of the industrial output indicators relating to gross domestic product (GDP).

The sectoral structure represents the composition, quantitative ratios, and forms of interconnection among industries and products; the degree of differentiation and specialization of these industries, and the specifics of economic relationships between them; and the qualitative state of the economy (Aliev et al. 2002).

Sectoral structures are presented both in classical economic literature and by modern researchers in various forms. Naturally, these structures have been expanded over time. The three-sector model in economics developed by Colin Clark divides economies into three sectors of activity: extraction of raw materials (primary), manufacturing (secondary), and service industries, which exist to facilitate the transport, distribution, and sale of goods produced in the secondary sector (tertiary) (Clark 1940). The five-sector model by Daniel Bell is supplemented with the service sector comprising transport, communications, and public utilities, trade, finance, insurance, real estate, healthcare, education, science, recreation, and public administration (Bell 1973). According to the five-sector model by Robert B. Reich, the third sector is specified as production services, the fourth is the sector of in-person services, and the fifth is the intellectual and communication services sector. The sectors are designated by the selection criteria for each group (Reich 1992). Some authors propose the structure of the economic system consisting of two groups. On the one hand, this is a group of production of goods and services; on the other hand, it is a group of the termed virtual production. The first group refers to the ideas of Robert B. Reich, while the second group of virtual production includes the so-called "ideas" along with virtual currency circulation and mass media activities (Lyubimtseva 2003). This group involves ideas for the technological updating of the material production of goods and services, rationalizing management and managerial work, and improving information support. Thus, it is urgent to solve the problem of calculating indicators having a significant impact on the sphere of material production (Vasin 2013). Addressing the issues of sectoral structures greatly simplifies the presence of various official classifiers worldwide, which are constantly being improved in terms of updating and the relevance of new industries and activities (Shirapov 2015). Some authors premise on the ten basic sectors of the Global Industry Classification Standard (GICS) used by Morgan Stanley Capital International (MSCI) and Standard & Poor's (S&P) (Taylor and Csomós 2012; Csomós 2013). However, for the purposes of our study, a larger grouping of inherently similar industries is preferable.

Coordinated development is associated with diverse gradual dynamics of socio-economic indicators. It is changes in the regional sectoral structure that mark the process of socio-economic dynamics.

Initially, the states have already determined their own specific structure estimated in the ratio between high-tech and low-tech manufacturing. This determines both the production capabilities of the state's economy and the effectiveness of institutional functioning (Constantine 2017). The priority of high-tech production in the structure of the economy enables sustainable economic growth (Andreoni and Scazzieri 2014; Hidalgo et al. 2007; Nelson and Winter 1990; Reinert 2008; Schumpeter 2008) due to economies of scale resulting in increasing returns. Conversely, with low-tech manufacturing, diminishing returns will denote a decline in the efficiency of production activities due to the rapid competitive increase.

Moreover, the indicators of the country's economy largely depend on the production structure that determines the rate of firm level innovation, diversification of economy, opportunities and diversity in the labor market, etc. Furthermore, institutional development is also associated with sectoral structures: either formal market institutions develop due to financial opportunities in the functioning of efficient industries, or there is an increase in nonmarket mechanisms viewed as corruption, shadow relations, political patronage, etc. in the absence of profitability of formal costs with a low-efficiency production structure (Constantine 2017).

When evaluating the sectoral structure efficiency of different territories, it is advisable to analyze the whole set of industries. It would be erroneous only to parallel the shares of a single type of production in cross-country or inter-regional comparisons. There are single-industry (Paci and Pigliaru 1997) and two-branch (Temple and Wößmann 2006) growth models. However, the latest research tends to analyze a wider range of industries, which is justified in terms of achieving territorial efficiency goals.

This is exemplified by the structure of Norway with the prevailing extraction of natural resources relating to industries with diminishing returns. Meanwhile, the country has been among the leaders in socio-economic development for many decades. Despite the devaluation of the national currency resulting from lower energy prices, the Norwegian economy is effective because of a high diversification of its production structure and an increase in the competitiveness of nonenergy export goods. However, even despite the efficient structure of production, "a decision was made to gradually sell energy assets in order to reduce dependence on fluctuations in energy prices" (Grigoryev et al. 2019, pp. 6–7), in which international organizations provide different project estimates for the growth rate of the country's economy. In particular, the Organisation for Economic Co-operation and Development (OECD) predicted a slowdown in growth rates due to a decrease in nonprimary exports (ibid., p. 14).

The question arises as to whether it is possible to change the existing structure and transform the economy toward increasing production returns.

Constantine (2017) noted that structural changes come from the two types of sources: state intervention and external shocks. However, the question arises if there are possible internal (intra-business) motives that serve as an incentive for business structures to diversify production, subjected to the factors of a competitive environment and affecting changes in the sectoral structure and labor market of the region and the state. If so, then all sources of industry dynamics can be divided into four groups:

1.  Planned changes caused by business expectations and interests;
2.  Structural changes caused by state intervention;
3.  Expected external and internal shocks: economic shocks (positive and negative), wars;
4.  Unpredicted sudden impacts: natural disasters, epidemics, pandemics.

We dwell on regional sectoral structural changes in Italy during the Italian "economic miracle" from 1951 to 1970 to consider systematic changes caused by both business expectations and interests, and political intervention. During this period, there was an intensive shift in the labor force from agricultural to the manufacturing, market service, and nonmarket service sectors, in which the role of the state was significant and especially noticeable in comparing the two periods. During the first period, the state stimulated industrialization with the help of a centralized supply-side top-down approach; in the second period, the approach of regional development policy changed toward demand-oriented measures, in particular, the provision of fiscal subsidies to firms, household income support, and public sector job creation (Piras 2022). The transition of the labor force between the agricultural and urban nonagricultural sectors is typically due to marginal product differences and intersectoral wage gaps (Temple and Wößmann 2006).

The example of Japan is also indicative in terms of the gradual change in the sectoral structure associated with various economic factors. In the 1970s, rising wage costs led to the reduction in the textile industry, and rising energy resources resulted in a decrease in the profitability of aluminum production. Instead, the production of automobiles,

electronics, and synthetic materials was developed in the 1980s. In turn, in China, since the 1970s, the heavy industry sector was accorded top-priority with high capital intensity, low employment, and value added. Industrial reorientation turned the priorities to light and food industries, bringing greater added value with a higher level of employment (Ishmuratov and Malganova 2013).

Various economic crises, in particular, the 2008 financial crisis, can serve as examples of sudden but predictable changes. Such periods are characterized by major changes in the nature of the socio-economic dynamics of states in comparison with the dynamics of coordinated development, and diversification of the behavior of economic entities. Said phenomena are viewed while analyzing territorial entities during major crises in dominant sectors in the total industry structure (Raźniak et al. 2017) and while searching for prerequisites for the sustainability of cities and key industries in a crisis situation (Raźniak et al. 2020).

The COVID-19 pandemic is a vivid example of unpredictable shocks.

The remainder of the paper is organized as follows. Section 3 presents the methodological framework and provides empirical support for our research. Section 4 describes the dataset used in this work and highlights the research results. In Section 5, we discuss the results obtained, summarize the main findings, and propose future research perspectives.

## 3. Data and Methodology

### 3.1. Methodological Approaches

We used the following approaches for the empirical analysis of the sectoral structure:

1. Static analysis. This is the basic position of the structure in an interstate comparison at a specific date;
2. Analysis of sectoral structure dynamics in a crisis;
3. Analysis of sectoral structure dynamics during stable (planned) growth.

In some cases, the 2nd and 3rd approaches could be combined into a single sequence of analysis.

### 3.2. Stages and Methods of Analysis

Methods for collecting and analyzing statistical data were applied in the context of each approach as follows:

1. Selection of statistical data group according to territorial and sectoral characteristics. Most of the European countries were selected for the study. The sample was limited by the availability of the necessary statistical data in the database of the statistical office of the European Union (Eurostat 2022a, 2022b). The list of countries, and their initial and calculated indicators required for the study are presented in Appendices A–C.

To conduct sectoral analysis, we used indicators characterizing the production of goods and services integrated into five sectors:

- Sector I: Mining and quarrying.
- Sector II: Manufacturing.
- Sector III: Electricity, gas, steam, and air conditioning supply; water supply; sewerage, waste management and remediation activities; wholesale and retail trade; repair of motor vehicles and motorcycles; transportation and storage.
- Sector IV: Real estate activities; accommodation and food service activities; administrative and support service activities.
- Sector V: Knowledge-intensive market services; information and communication; professional, scientific, and technical activities; information sector; computer-related services.

2. Data collection for a specific date, and grouping thereof on a territorial basis. Key indicator: the share of the industry in the total volume, in percentage. The calculation was premised on the total volume and absolute values of the industry, represented by the value added at factor cost, in million euro.

3. Time period choice for the analysis of sectoral dynamics during the economic crisis. We chose the period of 2008–2009 to analyze sectoral dynamics.
4. Data collection in dynamics for the period of critical impacts of sudden factors, and grouping thereof on a territorial basis (state).
5. Time period choice for the analysis of systematic dynamics of territorial development (lacking critical impacts). We chose a period of stable growth from 2010 to 2019 to exclude the influence of expected critical impacts and sudden unexpected events. The analysis resulted in the identified trends in the sectoral ratio that occur in a systematic manner.
6. Data collection in dynamics for the period of 2010–2019, and grouping thereof on a territorial basis (state).
7. Analysis of the indicator dynamics within the selected periods and deductions on the dynamics of territorial development. Structural shifts over a certain period were determined using the Gatev coefficient ($C_{Gat}$):

$$C_{Gat} = \sqrt{\frac{\sum (d_1 - d_0)^2}{\sum d_1^2 + \sum d_0^2}} \tag{1}$$

where $d_1$ is the share of the indicator at the end of the period; $d_0$ is the share of the indicator at the beginning of the period.

In our opinion, this coefficient is optimal and illustrative for this type of calculation and is devoid of some shortcomings characteristic of other indicators. In particular, in contrast to the linear and quadratic coefficient of absolute structural shifts, the values of the Gatev coefficient vary from 0 to 1, and approaching 1 indicates notable differences in structural shifts. An alternative to the Gatev coefficient is the Salai coefficient, but its value alters greatly with the changes in values of the elements the totality is divided into. An application of various coefficients in calculations, and their advantages and disadvantages are clearly shown in regional studies of sectoral structures (Trifonov and Veselova 2015).

A visual interpretation of statistical information was implemented with a graphical analysis of sectoral dynamics using techniques for constructing linear and polynomial trends. The tightness of the connection between sectoral changes was determined using a correlation analysis by comparing absolute data of sectoral dynamics indicators. The sectoral dynamics of different countries was compared using a graphical method via the construction of graphs and normalized stacked bar charts.

8. Judgments on territorial sectoral transformation.

As the periods of crisis and stable growth follow each other in our research, in some cases, they were concomitantly analyzed, with the results being considered in terms of comparison and mutual effect.

## 4. Results

An analysis of the sectoral structure in European countries in the period from 2008 to 2019 has showed mainly a loss in value added in the mining and quarrying industry, and an increase in the share of the manufacturing industry.

Thus, there is a decrease in the share of the mining and quarrying sector in 21 out of 25 analyzed countries, with major effects in 14 states (Table 1). There is just an increase in production in four countries. The Gatev coefficient indicates significant changes in the dynamics of the mining and quarrying industry in Romania, Denmark, Czechia, Italy, Ireland, Germany, France, the Netherlands, and Cyprus. Major changes have taken place in Norway, Spain, Austria, Poland, etc. European countries continue reorienting toward manufacturing industries and the service sector.

**Table 1.** Calculation of the Gatev coefficient for the mining and quarrying industry for 2008–2019. Source: own elaboration based on Eurostat (2022a).

|  | $d_0$ | $d_1$ | $(d_1 - d_0)^2$ | $d_0{}^2$ | $d_1{}^2$ | $(d_1 + d_0)^2$ | $C_{Gat}$ | |
|---|---|---|---|---|---|---|---|---|
| Bulgaria | 1.50 | 1.23 | 0.073359 | 2.247745 | 1.508963 | 7.440056 | 0.14 | ↓ |
| Czechia | 1.48 | 0.46 | 1.047083 | 2.201818 | 0.212135 | 3.780823 | 0.66 | ↓ |
| Denmark | 4.12 | 0.90 | 10.37231 | 16.98081 | 0.810307 | 25.20992 | 0.76 | ↓ |
| Germany | 0.30 | 0.12 | 0.031251 | 0.087332 | 0.0141 | 0.171613 | 0.56 | ↓ |
| Estonia | 0.77 | 0.76 | 0.000203 | 0.593898 | 0.572145 | 2.331883 | 0.01 | ↓ |
| Ireland | 0.36 | 0.14 | 0.049233 | 0.131108 | 0.019657 | 0.252296 | 0.57 | ↓ |
| Greece | 0.17 | 0.19 | 0.000615 | 0.027908 | 0.03681 | 0.128821 | 0.10 | ↑ |
| Spain | 0.24 | 0.12 | 0.013854 | 0.058485 | 0.015409 | 0.133932 | 0.43 | ↓ |
| France | 0.15 | 0.06 | 0.008455 | 0.02357 | 0.003791 | 0.046267 | 0.56 | ↓ |
| Italy | 0.40 | 0.15 | 0.060496 | 0.157568 | 0.022798 | 0.300235 | 0.58 | ↓ |
| Cyprus | 0.36 | 0.11 | 0.061623 | 0.130922 | 0.012903 | 0.226026 | 0.65 | ↓ |
| Latvia | 0.27 | 0.42 | 0.02354 | 0.072053 | 0.177962 | 0.476491 | 0.31 | ↑ |
| Lithunania | 0.38 | 0.23 | 0.024018 | 0.14808 | 0.052825 | 0.377792 | 0.35 | ↓ |
| Luxemburg | 0.09 | 0.07 | 0.000556 | 0.008001 | 0.00434 | 0.024127 | 0.21 | ↓ |
| Hungary | 0.23 | 0.26 | 0.001204 | 0.052914 | 0.070084 | 0.244792 | 0.10 | ↑ |
| Netherlands | 1.54 | 0.67 | 0.750741 | 2.360717 | 0.448913 | 4.868519 | 0.52 | ↓ |
| Austria | 0.44 | 0.26 | 0.032448 | 0.192319 | 0.066775 | 0.48574 | 0.35 | ↓ |
| Poland | 2.74 | 1.61 | 1.267959 | 7.48341 | 2.590636 | 18.88013 | 0.35 | ↓ |
| Portugal | 0.33 | 0.25 | 0.006108 | 0.106788 | 0.061817 | 0.331102 | 0.19 | ↓ |
| Romania | 3.03 | 0.62 | 5.799461 | 9.180251 | 0.386493 | 13.33403 | 0.78 | ↓ |
| Slovenia | 0.41 | 0.31 | 0.009368 | 0.165057 | 0.09578 | 0.512308 | 0.19 | ↓ |
| Slovakia | 0.52 | 0.35 | 0.028192 | 0.271361 | 0.124621 | 0.763772 | 0.27 | ↓ |
| Finland | 0.23 | 0.35 | 0.014961 | 0.054128 | 0.126004 | 0.345303 | 0.29 | ↑ |
| Sweden | 0.62 | 0.48 | 0.019352 | 0.382931 | 0.230115 | 1.20674 | 0.18 | ↓ |
| Norway | 28.15 | 15.71 | 154.7713 | 792.5343 | 246.8442 | 1923.986 | 0.39 | ↓ |

The graphs showing the dynamics of changes in the share of the mining and quarrying industry in the European countries with the largest production volume are presented in Figure 1. Changes for all analyzed states based on the submitted primary data are shown in Appendix A.

An analysis of the dynamics has revealed a general trend of the decline in the level of the mining and quarrying industry in the European countries. This is evidenced by linear trends built on the basis of statistical data.

However, the decline is not steady. The fifth-degree polynomial exponential curves vividly express the decline in the crisis of 2008–2009 with a subsequent recovery in production, and a further gradual drop to below the 2008 levels in 2019. However, a sharp post-crisis recovery in mining, often being higher than the pre-crisis level, is related to a natural increase in demand and the restoration of lending opportunities. Apparently, the aggregate demand was made up of the volumes of current consumption and replenishment of stocks, probably being used during the crisis. In addition, there is some rise in the share of the mining and quarrying industry in the sectoral structure in 2018, which is due to an increase in prices rather than a sharp production growth.

On the contrary, there is growth as a share of GVA in the manufacturing industry. As the given dynamics is less intense than the dynamics of changes in the share of the mining and quarrying industry, we analyze the crisis period of 2008–2009 (Table 2), and the period of stable growth of 2010–2019 (Table 3).

According to Table 2, there is a negative trend in the share of the manufacturing industry in almost all countries during the crisis period from 2008 to 2009. This is mainly due to a drop in demand and credit problems. However, the largest decline, as shown by the Gatev coefficient, was recorded in Luxembourg, Finland, Spain, and Slovakia.

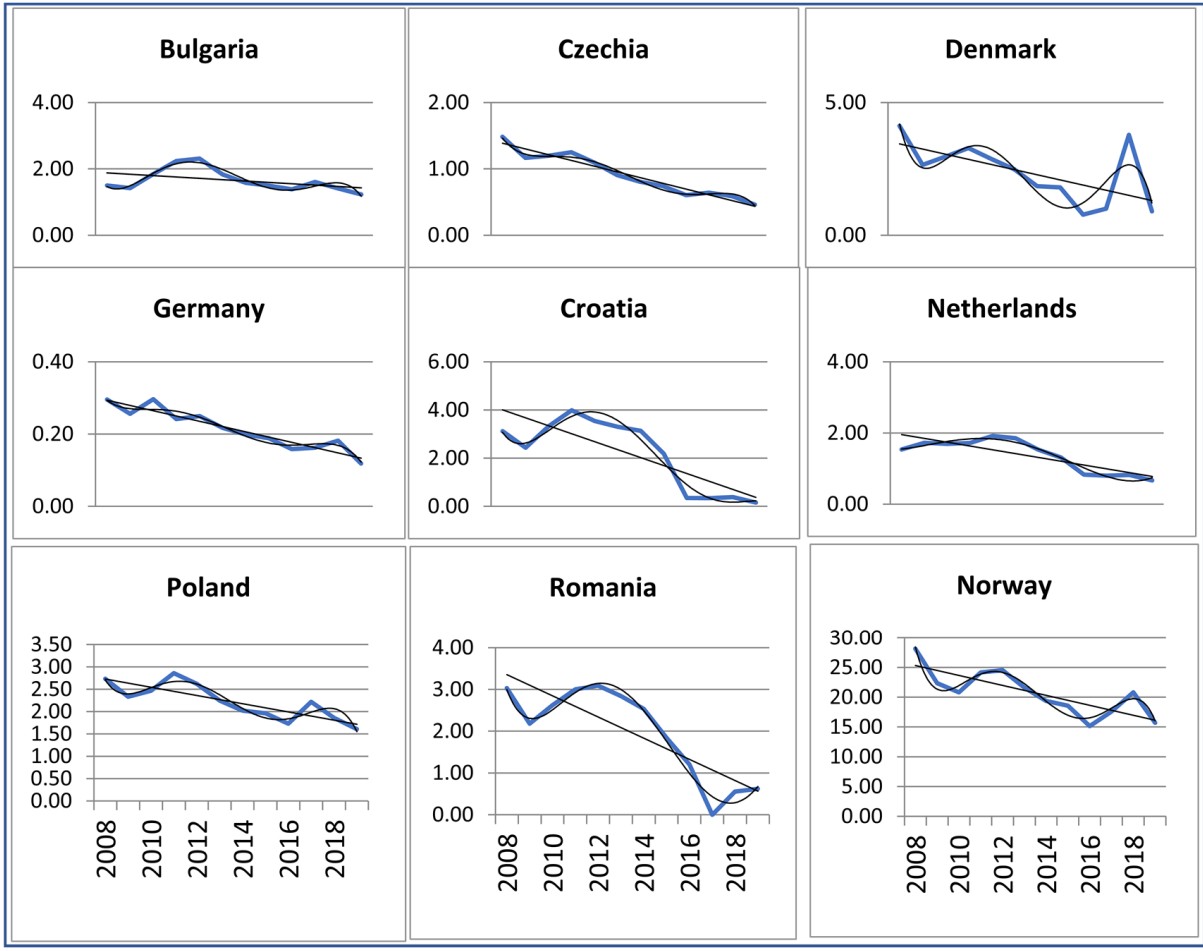

**Figure 1.** Dynamics of mining and quarrying industry share in the structure of gross value added (GVA) at factor cost for 2008–2019, in % of total GVA. Source: own elaboration based on Eurostat (2022a).

**Table 2.** Calculation of the Gatev coefficient for the manufacturing industry for 2008–2009. Source: own elaboration based on Eurostat (2022b).

| | $d_0$ | $d_1$ | $(d_1 - d_0)^2$ | $d_0^2$ | $d_1^2$ | $(d_1 + d_0)^2$ | $C_{Gat}$ | |
|---|---|---|---|---|---|---|---|---|
| Bulgaria | 13.66 | 11.94 | 2.950376 | 186.6196 | 142.6403 | 655.5696 | 0.09 | ↓ |
| Czechia | 21.58 | 19.34 | 5.035438 | 465.8047 | 373.9787 | 1674.531 | 0.08 | ↓ |
| Denmark | 13.84 | 12.46 | 1.898609 | 191.4714 | 155.2372 | 691.5186 | 0.07 | ↓ |
| Belgium | 15.66 | 14.46 | 1.439332 | 245.1369 | 209.0085 | 906.8515 | 0.06 | ↓ |
| Germany | 19.82 | 17.40 | 5.85546 | 392.8147 | 302.7512 | 1385.276 | 0.09 | ↓ |
| Estonia | 14.53 | 12.90 | 2.660762 | 211.0901 | 166.3521 | 752.2236 | 0.08 | ↓ |
| Ireland | 20.83 | 21.27 | 0.191323 | 433.8645 | 452.2776 | 1772.093 | 0.01 | ↑ |
| Greece | 7.97 | 7.96 | 0.000151 | 63.51891 | 63.32326 | 253.6842 | 0.00 | ↓ |
| Spain | 12.39 | 10.06 | 5.42473 | 153.5366 | 101.2415 | 504.1315 | 0.15 | ↓ |
| France | 11.34 | 10.31 | 1.053693 | 128.5341 | 106.3125 | 468.6396 | 0.07 | ↓ |
| Croatia | 14.92 | 13.30 | 2.647902 | 222.7061 | 176.7864 | 796.337 | 0.08 | ↓ |
| Italy | 14.33 | 12.65 | 2.840278 | 205.4498 | 159.9771 | 728.0136 | 0.09 | ↓ |
| Cyprus | 7.48 | 7.24 | 0.058532 | 55.96369 | 52.40245 | 216.6737 | 0.02 | ↓ |
| Latvia | 8.49 | 7.22 | 1.602198 | 72.07419 | 52.18433 | 246.9149 | 0.11 | ↓ |
| Lithunania | 9.24 | 8.98 | 0.069457 | 85.41409 | 80.61216 | 331.9831 | 0.02 | ↓ |

**Table 2.** *Cont.*

| | $d_0$ | $d_1$ | $(d_1 - d_0)^2$ | $d_0{}^2$ | $d_1{}^2$ | $(d_1 + d_0)^2$ | $C_{Gat}$ | |
|---|---|---|---|---|---|---|---|---|
| Luxemburg | 8.29 | 5.66 | 6.889863 | 68.65691 | 32.04796 | 194.5199 | 0.26 | ↓ |
| Hungary | 20.76 | 19.20 | 2.445387 | 431.0359 | 368.549 | 1596.724 | 0.06 | ↓ |
| Netherlands | 10.26 | 9.43 | 0.675946 | 105.1651 | 88.97853 | 387.6113 | 0.06 | ↓ |
| Austria | 17.80 | 16.06 | 3.021628 | 316.7378 | 257.8865 | 1146.227 | 0.07 | ↓ |
| Poland | 17.92 | 16.25 | 2.779406 | 320.9763 | 264.0188 | 1167.211 | 0.07 | ↓ |
| Portugal | 11.99 | 10.77 | 1.499338 | 143.8758 | 116.0004 | 518.253 | 0.08 | ↓ |
| Romania | 11.78 | 10.06 | 2.959616 | 138.819 | 101.2397 | 477.1578 | 0.11 | ↓ |
| Slovenia | 20.31 | 16.77 | 12.51678 | 412.4602 | 281.2735 | 1374.951 | 0.13 | ↓ |
| Slovakia | 13.38 | 10.83 | 6.509648 | 178.9882 | 117.2293 | 585.9254 | 0,15 | ↓ |
| Finland | 18.77 | 14.28 | 20.18457 | 352.3933 | 203.9018 | 1092.406 | 0.19 | ↓ |
| Sweden | 15.99 | 14.09 | 3.620525 | 255.8149 | 198.5689 | 905.1472 | 0.09 | ↓ |
| Norway | 8.32 | 7.72 | 0.359956 | 69.227 | 59.60323 | 257.3005 | 0.05 | ↓ |

**Table 3.** Calculation of the Gatev coefficient for the manufacturing industry for 2010–2019. Source: own elaboration based on Eurostat (2022b).

| | $d_0$ | $d_1$ | $(d_1 - d_0)^2$ | $d_0{}^2$ | $d_1{}^2$ | $(d_1 + d_0)^2$ | $C_{Gat}$ | |
|---|---|---|---|---|---|---|---|---|
| Bulgaria | 12.34 | 15.68 | 11.18568 | 152.2621 | 245.9864 | 785.3114 | 0.17 | ↑ |
| Czechia | 21.52 | 21.57 | 0.00272 | 462.9697 | 465.2167 | 1856.37 | 0.00 | ↑ |
| Denmark | 14.77 | 15.24 | 0.223058 | 218.0568 | 232.2282 | 900.347 | 0.02 | ↑ |
| Belgium | 19.77 | 14.88 | 23.93418 | 390.7748 | 221.2885 | 1200.192 | 0.20 | ↓ |
| Germany | 14.76 | 21.04 | 39.45148 | 217.8837 | 442.7627 | 1281.841 | 0.24 | ↑ |
| Estonia | 22.61 | 14.21 | 70.47417 | 511.1571 | 202.0348 | 1355.909 | 0.31 | ↓ |
| Ireland | 8.03 | 32.97 | 622.1624 | 64.44472 | 1087.082 | 1680.892 | 0.74 | ↑ |
| Greece | 10.77 | 7.53 | 10.52329 | 116.0307 | 56.66769 | 334.8736 | 0.25 | ↓ |
| Spain | 10.79 | 11.12 | 0.108006 | 116.3607 | 123.5589 | 479.7311 | 0.02 | ↑ |
| France | 12.79 | 11.57 | 1.475763 | 163.482 | 133.8926 | 593.2734 | 0.07 | ↓ |
| Croatia | 14.18 | 15.32 | 1.279013 | 201.19 | 234.5516 | 870.2042 | 0.05 | ↑ |
| Italy | 6.77 | 15.54 | 76.97044 | 45.83293 | 241.5936 | 497.8827 | 0.52 | ↑ |
| Cyprus | 9.65 | 6.54 | 9.650671 | 93.09709 | 42.79947 | 262.1424 | 0.27 | ↓ |
| Latvia | 9.94 | 10.58 | 0.402601 | 98.87769 | 111.899 | 421.1509 | 0.04 | ↑ |
| Lithunania | 6.23 | 12.60 | 40.5592 | 38.85539 | 158.8109 | 354.7734 | 0.45 | ↑ |
| Luxemburg | 20.65 | 5.40 | 232.5079 | 426.2531 | 29.13506 | 678.2685 | 0.71 | ↓ |
| Netherlands | 17.12 | 10.69 | 41.3362 | 293.1651 | 114.3347 | 773.6635 | 0.32 | ↓ |
| Austria | 15.53 | 17.55 | 4.073901 | 241.0801 | 307.8321 | 1093.75 | 0.09 | ↑ |
| Poland | 11.41 | 18.29 | 47.45665 | 130.0831 | 334.6806 | 882.0708 | 0.32 | ↑ |
| Portugal | 11.41 | 12.14 | 0.535514 | 130.2222 | 147.4593 | 554.8277 | 0.04 | ↑ |
| Romania | 19.53 | 11.62 | 62.48617 | 381.3173 | 135.0832 | 970.3148 | 0.35 | ↓ |
| Slovenia | 15.62 | 21.65 | 36.3464 | 244.1386 | 468.8841 | 1389.699 | 0.23 | ↑ |
| Slovakia | 16.10 | 17.22 | 1.262687 | 259.0997 | 296.5376 | 1110.012 | 0.05 | ↑ |
| Finland | 15.37 | 13.81 | 2.418091 | 236.0978 | 190.7286 | 851.2349 | 0.08 | ↓ |
| Norway | 16.74 | 6.57 | 103.3142 | 280.1533 | 43.20987 | 543.4121 | 0.57 | ↓ |

Based on Table 3, there is a steady increase in the share of the manufacturing industry in such countries as Ireland, Italy, Lithuania, Poland, Slovenia, and Germany during 2010–2019. There was a decrease in the share of mining and quarrying industry in most countries during the same period. This is largely due to the political decisions of governmental authorities to reduce mining volume. The development of manufacturing enterprises as producers of goods with a greater added value provided higher guarantees for the stability of the entire socio-economic system and helped to accelerate the recovery from the recession.

Correlation coefficients showing the feedback tightness between the dynamics of the absolute indicators of mining and manufacturing industries are presented in Table 4. Absolute indicators provide more reliable data in correlation than relative ones, as the latter are initially interconnected with each other.

**Table 4.** Correlation between the volume dynamics of mining and manufacturing industries (value added at factor cost, in EUR million) for 2008–2019, $p < 0.05$. Source: own elaboration based on Eurostat (2022b).

|  | $r$ | $t$-Test |
| --- | --- | --- |
| Czechia | −0.6995 | 5.4976 |
| Germany | −0.702 | 7.2902 |
| France | −0.84 | 7.0469 |
| Croatia | −0.90 | 7.7919 |
| Italy | −0.765 | 5.3466 |
| Netherlands | −0.874 | 8.3166 |
| Romania | −0.755 | 6.3407 |

As we see, there is a strong inverse relationship between mining and quarrying output and the production of manufacturing industries in a number of countries. However, there is no significant correlation regarding many European countries in spite of tendencies of opposite changes in mining and manufacturing industries. In addition, there is a positive correlation concerning two countries, including Norway as the leading mineral producer. The results of the correlation analysis of all studied countries are presented in Table A2 in Appendix B.

Overall, it can be concluded that there is a trend toward an increase in the volume of manufacturing industries with a decrease in the level of mining and quarrying. As a rule, such a replacement occurs gradually. However, it should be recognized that higher value-added industries, which are usually manufacturing ones, provide greater independence and stability than depleting natural resources.

These results are supported by the ratio between the volume of mining and the volume of services (Table 5). A significant inverse correlation is observed in the countries with a relationship between the volumes of mining and manufacturing industries: Germany, France, Italy, the Netherlands, and Romania. Table A3 in Appendix B presents the results of the related correlation analysis and other analyzed states.

**Table 5.** Correlation between the volume dynamics of mining industry and services in the total (value added at factor cost, in million euro) for 2008–2019, $p < 0.05$. Source: own elaboration based on Eurostat (2022b).

|  | $r$ | $t$-Test |
| --- | --- | --- |
| Germany | −0.826 | 7.6528 |
| France | −0.862 | 6.5121 |
| Croatia | −0.718 | 3.5611 |
| Italy | −0.888 | 6.8903 |
| Cyprus | −0.5798 | 2.9366 |
| Netherlands | −0.8798 | 7.6160 |
| Romania | −0.835 | 7.7867 |

Let us carry out a comparative analysis of sectoral dynamics in the production of goods and services in the sectoral structure of the European countries for 2008–2009 and 2010–2019 (Figure 2). Primary data are given in Appendix C.

In most European countries, the 2008 economic crisis was accompanied by a decrease in the share of production of goods and services in Sectors I–III. These sector-specific industries were the most affected by the economic environment, and the countries focused mainly on these industries found the worst position. The largest decline in the share of production in Sectors I–III was observed in Hungary (by 2.22%), Slovenia (by 2.62%), and Denmark (by 7.39%) in 2009 compared to 2008.

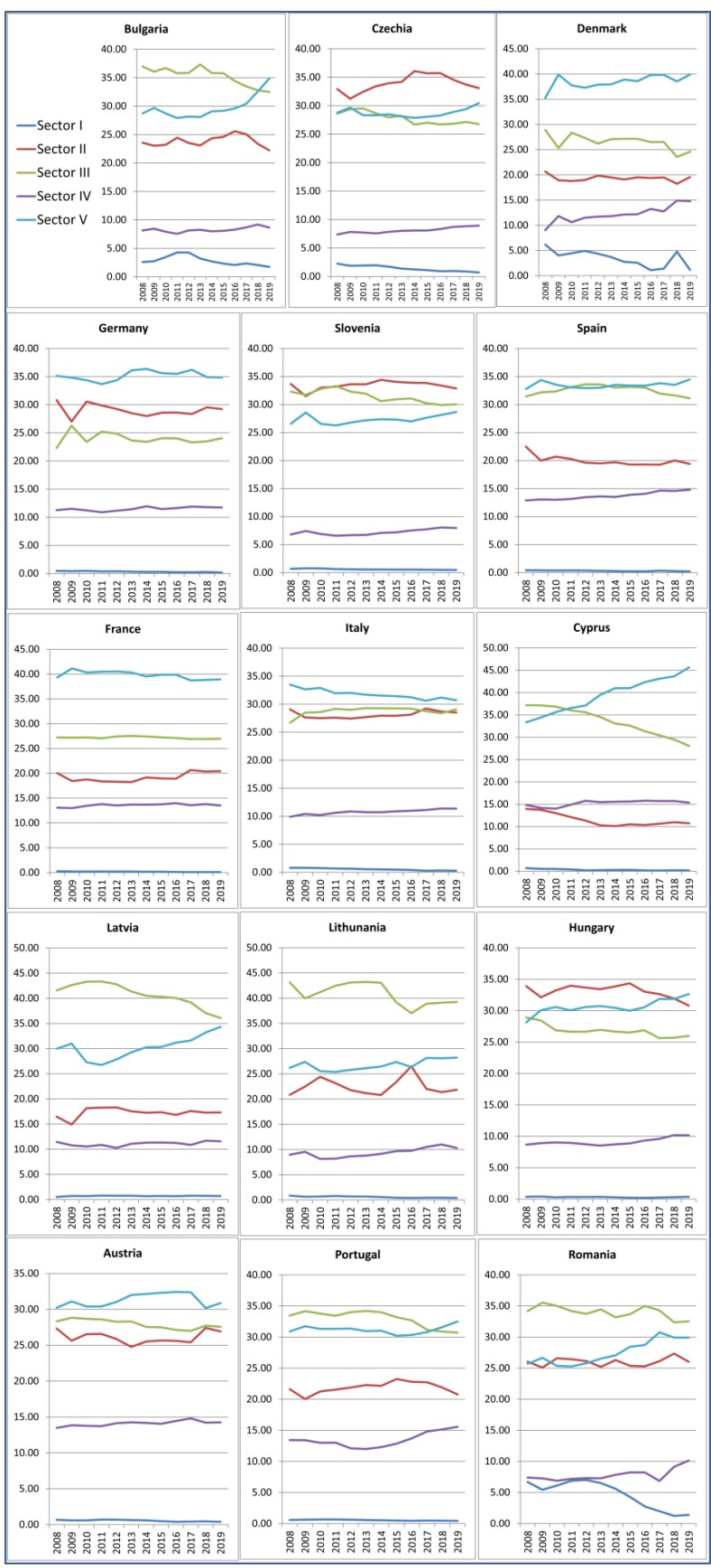

**Figure 2.** Dynamics of sector indicators specific for the production of goods and services for 2008–2019, in %. Source: own elaboration based on Eurostat (2022a).



However, the period of 2008–2009 was characterized by a predominant increase in the share of Sector IV and, especially, Sector V industries. This was especially pronounced in Denmark and Bulgaria (by 1.25%), Czechia (by 1.33%), and Slovenia and Lithuania (by 1.76%).

Figure 3 clearly illustrates the shares of the above sectors in the European countries by years. There is an evident trend of increasing the share of Sector V during the period of stable growth of 2010–2019 after the crisis of 2008–2009: the share of intellectual and communication services increased by 6.18% in Bulgaria, by 10.36% in Cyprus, by 7.04% in Latvia, by 4.53% in Romania, by 2.83% in Lithuania, by 2.11% in Slovenia, by 2.07% in Hungary, and by 1.18% in Portugal. An increase in the share of knowledge-intensive services is due to a decrease in the volume of mining and quarrying, and a decrease in the share of manufacturing services. At the same time, it should be noted that, in general, the service sectors are less demanding on the availability of a permanent source of credit, which reduced their dependence on shocks in the financial sector and allowed both to enter a positive development trend and to somewhat compensate for the countries' losses from the decline in the manufacturing and especially the mining industry.

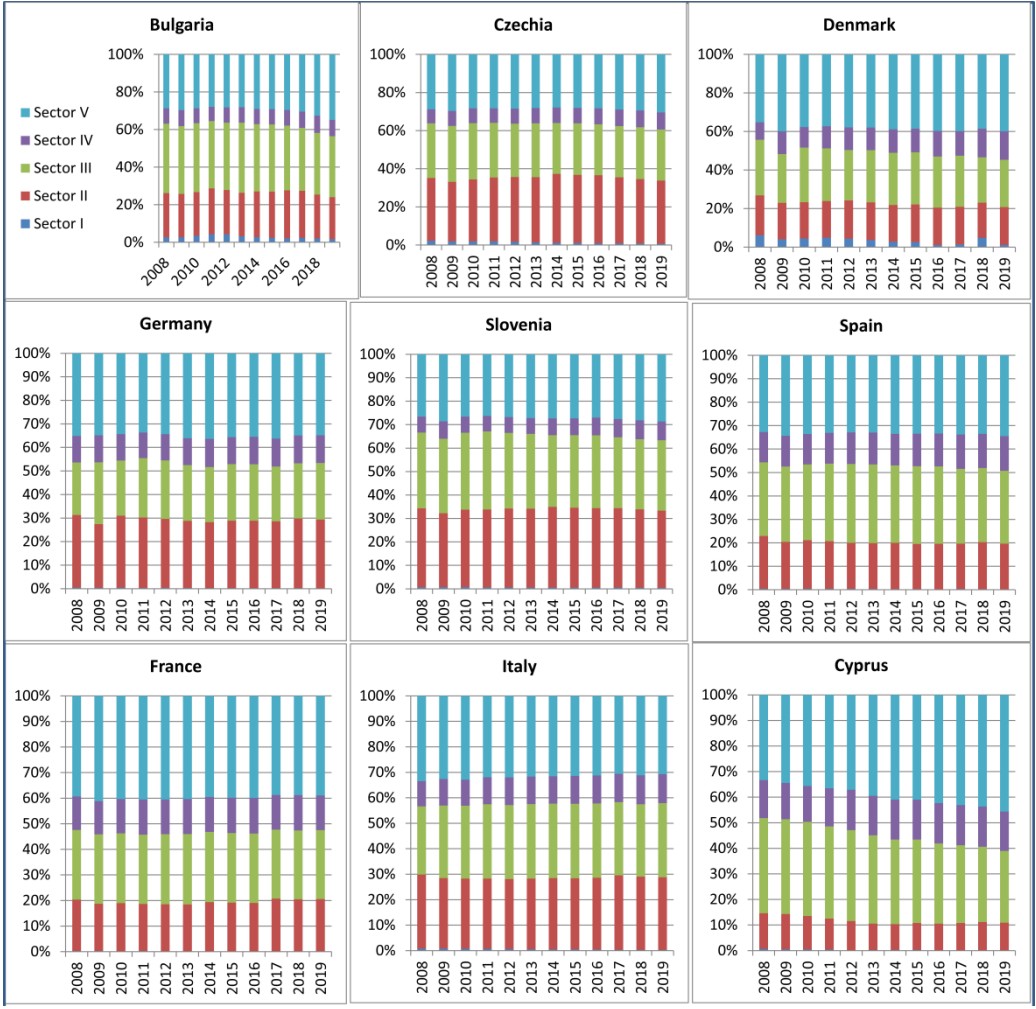

**Figure 3.** *Cont.*

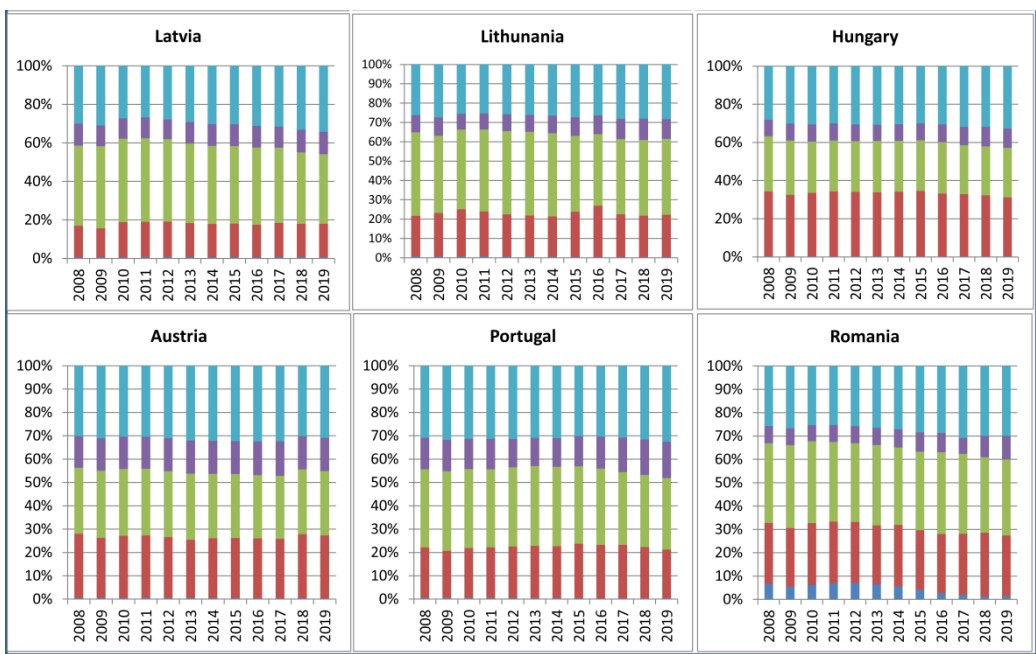

**Figure 3.** Ratio of the share in the sectors specific for the production of goods and services for 2008–2019, in %. Source: own elaboration based on Eurostat (2022a).

A vivid example is Norway, which, being the leader in oil production in Europe, reduced the share of mining and quarrying by 7.13% during 2009–2019, in which the share of industries in Sectors IV and V increased by 7.57% (Table 6).

**Table 6.** Ratio of sectors specific for production of goods and services in Norway for 2009–2019, in %. Source: own elaboration based on Eurostat (2022a).

| Norway | 2009 | 2010 | 2011 | 2012 | 2013 | 2014 | 2015 | 2016 | 2017 | 2018 | 2019 |
|---|---|---|---|---|---|---|---|---|---|---|---|
| Sector I | 30.35 | 29.03 | 33.23 | 33.41 | 30.04 | 27.52 | 26.19 | 22.20 | 25.58 | 30.03 | 23.22 |
| Sector II | 10.48 | 10.76 | 9.94 | 9.62 | 9.76 | 10.40 | 10.48 | 10.48 | 9.95 | 9.07 | 9.72 |
| Sector III | 21.42 | 21.70 | 19.50 | 19.13 | 20.58 | 20.92 | 21.94 | 23.50 | 21.92 | 20.52 | 21.74 |
| Sector IV | 8.82 | 8.89 | 8.89 | 8.99 | 9.18 | 9.60 | 9.78 | 10.75 | 10.63 | 10.08 | 10.94 |
| Sector V | 28.94 | 29.63 | 28.44 | 28.85 | 30.44 | 31.55 | 31.61 | 33.06 | 31.92 | 30.29 | 34.38 |

## 5. Discussion and Conclusions

In this paper, we have raised issues concerning the correlation between the sectors of production of goods and services in the European countries, affecting both the effective functioning of the socio-economic system during the stabilization period and successful counteracting crisis shock factors.

It has been noted that the sectors of information and communication services are the least affected during the economic crisis. The manufacturing sector, in particular, the mining and quarrying industry, is less resilient because of a sharp decline in prices. Yet, manufacturing industries also experience a decline in productivity.

The information and communication services sector has evidenced the greatest growth during the past 10 years of stability.

Thus, it is expedient to develop programs for the advancement of these industries, especially in tandem with the sectors of traditional and innovative production of goods.

It should be noted that the revealed changes in the structure of industries during the crisis and the stable growth periods are the symbols of the fourth industrial revolution (Schwab 2016). The emergence of this concept has been evidenced not merely from statistical sources but the modern society actually being faced with an active growth of Sector V industries. Undoubtedly, advanced smart technologies are a major benefit for the state

and business (Müller et al. 2018; Lopes de Sousa Jabbour et al. 2018; Carvalho et al. 2018). However, the researchers have revealed a number of challenges entailed by the fourth industrial revolution. For instance, it is emigration and immigration (McKenzie 2017), fear of total control, job loss, etc. (Caruso 2018). A major social challenge and a danger of the fourth industrial revolution is related to an increase in the annual demand for highly qualified developers of the emerging technologies to the detriment of lower-qualification employees (Balatsky 2019).

This paper has examined a case study of some European countries being quite attractive for highly qualified personnel in the information and communication service sectors. It is evident that the pace of high-tech industrial development is higher in the countries with a relatively high standard of living. However, there are some limitations of our research. Its outcomes cannot be unambiguously applied either on a global scale or to any state. Within the framework of our research, we have shown that every European country is not subject to the specific characteristics of the studied economic dynamics. It seems that even greater differences in economic behavior will be revealed in the analysis of transformations in similar periods in less developed and backward countries. In future research, we plan to analyze the statistics of these countries, and to study the migration flows of highly qualified personnel.

There is another subject area regarding sectoral dynamics to be investigated. Currently, there are a lot of debates related to the efficiency of the economic sectoral structure. On the one hand, some disputers advocate the ideas of total world globalization and, accordingly, the narrow specialization of a particular territory; on the other hand, the others advocate a full cycle of production within one region or state. Some are trying to prove that a high level of economic security is possible only in the presence of strategic natural resources, while the others pragmatically defend the thesis of rich mineral resources as being significant trouble for the state. It should be noted that different concepts of economic advantages may have priority at certain stages of history. However, this is the weakness of either concept as advantages sometimes turn into disadvantages, leading to a bifurcation. There is either a completely new model of the sectoral structure or a return to the previous one in the light of the revealed problems.

A striking example of a sudden factor emergence having had a shock effect on the global economy is a pandemic due to COVID-19 outbreak (World Health Organization 2020). One of the most important reactions of the countries at the virus epicenter was the closure of borders. This decision would likely only affect these countries if production were localized exclusively within the borders thereof. However, world globalization and the desire to reduce the cost of production have made economies of scale almost critical in making decisions on the distributed nature of production and building global value chains. Undoubtedly, economies of scale have affected the global competitive environment to further strengthen global approaches to economic activity. As a result, a number of countries having decided to suspend their production knocked out certain links from the global value chains, which led to an overstocking of intermediate products in those industries that had existed before the "knocked-out" link and the stoppage of the rest. This affected the output volume of final products and, consequently, the structure of the goods offered on the market.

Indeed, there would be no particular change on a global scale if these were only a few cases. However, as the pandemic has seriously affected most developed countries, they have revised this concept of organizing production.

China was one of the first to react with its own ideas and plans to change the concept of production. China actually turned out to be more prepared for the emergence of such an unpredictable factor as a pandemic than many countries did, as it had already taken certain actions in terms of revealing potential shortcomings in the existing world order. In fact, the pandemic has only accelerated the transformation of China's manufacturing model toward stimulating domestic demand, giving rise to a "dual circulation" strategy (Lukonin and Zakliazminskaia 2020). Transformation in the sectoral structure as such will

occur through the shifting of a key producer during the same production cycle to move as many links in the value chain as possible within the country borders. Even though China's previous functions were to assemble a finished product from imported components, modern production is focused on the same components via the domestic market.

However, we are talking about changes in the sectoral structure and the structure of global production, rather than about full rejection of international relations in terms of supplying certain links in the global chain. If China, focusing on import substitution with its own production, is currently unprepared to completely close the entire cycle within its borders, it will inevitably affect the quality of finished products.

This is rather about reglobalization than deglobalization (Smorodinskaya and Katukov 2020), although the bias toward the latter is quite noticeable. These issues also require special research.

**Funding:** This research was funded by the Russian Science Foundation, grant number 22-28-01976. Available online: https://rscf.ru/project/22-28-01976/ (accessed on 12 June 2022).

**Institutional Review Board Statement:** Not applicable.

**Informed Consent Statement:** Not applicable.

**Data Availability Statement:** Data that support the findings of this study are openly available via Eurostat Data Browser at https://ec.europa.eu/eurostat/web/main/data/database (accessed on 20 April 2022).

**Conflicts of Interest:** The author declares no conflict of interest.

## Appendix A

**Table A1.** Dynamics of mining and quarrying production share in the structure of gross value added (GVA) at factor cost for 2008–2019, in % of total GVA. Source: own elaboration based on Eurostat (2022a).

|  | 2008 | 2009 | 2010 | 2011 | 2012 | 2013 | 2014 | 2015 | 2016 | 2017 | 2018 | 2019 |
|---|---|---|---|---|---|---|---|---|---|---|---|---|
| Bulgaria | 1.50 | 1.42 | 1.84 | 2.24 | 2.31 | 1.83 | 1.58 | 1.49 | 1.38 | 1.61 | 1.41 | 1.23 |
| Czechia | 1.48 | 1.17 | 1.20 | 1.25 | 1.10 | 0.91 | 0.80 | 0.74 | 0.61 | 0.64 | 0.59 | 0.46 |
| Denmark | 4.12 | 2.65 | 2.96 | 3.29 | 2.88 | 2.48 | 1.85 | 1.80 | 0.78 | 1.00 | 3.78 | 0.90 |
| Germany | 0.30 | 0.26 | 0.30 | 0.24 | 0.25 | 0.22 | 0.20 | 0.19 | 0.16 | 0.16 | 0.18 | 0.12 |
| Estonia | 0.77 | 0.97 | 1.08 | 1.06 | 0.98 | 1.27 | 1.22 | 1.05 | 0.95 | 1.04 | 0.91 | 0.76 |
| Ireland | 0.36 | 0.30 | 0.36 | 0.38 | 0.34 | 0.32 | 0.28 | 0.11 | 0.23 | 0.28 | 0.18 | 0.14 |
| Greece | 0.17 | 0.16 | 0.18 | 0.18 | 0.15 | 0.19 | 0.22 | 0.23 | 0.27 | 0.51 | 0.29 | 0.19 |
| Spain | 0.24 | 0.19 | 0.20 | 0.20 | 0.19 | 0.16 | 0.15 | 0.13 | 0.13 | 0.19 | 0.15 | 0.12 |
| France | 0.15 | 0.13 | 0.12 | 0.14 | 0.12 | 0.11 | 0.10 | 0.09 | 0.07 | 0.06 | 0.05 | 0.06 |
| Italy | 0.40 | 0.36 | 0.39 | 0.34 | 0.32 | 0.27 | 0.25 | 0.24 | 0.22 | 0.15 | 0.17 | 0.15 |
| Cyprus | 0.36 | 0.28 | 0.26 | 0.20 | 0.10 | 0.12 | 0.14 | 0.17 | 0.10 | 0.08 | 0.11 | 0.11 |
| Latvia | 0.27 | 0.34 | 0.37 | 0.40 | 0.40 | 0.40 | 0.37 | 0.39 | 0.39 | 0.43 | 0.44 | 0.42 |
| Lithunania | 0.38 | 0.26 | 0.28 | 0.36 | 0.30 | 0.30 | 0.27 | 0.22 | 0.23 | 0.24 | 0.25 | 0.23 |
| Luxemburg | 0.09 | 0.09 | 0.09 | 0.09 | 0.08 | 0.07 | 0.06 | 0.05 | 0.06 | 0.06 | 0.06 | 0.07 |
| Hungary | 0.23 | 0.25 | 0.18 | 0.22 | 0.22 | 0.23 | 0.19 | 0.16 | 0.14 | 0.18 | 0.23 | 0.26 |
| Netherlands | 1.54 | 1.72 | 1.69 | 1.72 | 1.92 | 1.85 | 1.53 | 1.31 | 0.83 | 0.80 | 0.82 | 0.67 |
| Austria | 0.44 | 0.37 | 0.39 | 0.47 | 0.46 | 0.43 | 0.39 | 0.32 | 0.26 | 0.29 | 0.30 | 0.26 |
| Poland | 2.74 | 2.33 | 2.47 | 2.86 | 2.62 | 2.25 | 2.02 | 1.96 | 1.73 | 2.22 | 1.86 | 1.61 |
| Portugal | 0.33 | 0.34 | 0.35 | 0.34 | 0.31 | 0.28 | 0.27 | 0.25 | 0.24 | 0.27 | 0.26 | 0.25 |
| Romania | 3.03 | 2.18 | 2.62 | 3.00 | 3.09 | 2.84 | 2.53 | 1.84 | 1.20 | 0.00 | 0.55 | 0.62 |
| Slovenia | 0.41 | 0.41 | 0.45 | 0.37 | 0.35 | 0.33 | 0.33 | 0.33 | 0.34 | 0.34 | 0.32 | 0.31 |
| Slovakia | 0.52 | 0.55 | 0.48 | 0.50 | 0.47 | 0.47 | 0.46 | 0.48 | 0.40 | 0.39 | 0.40 | 0.35 |
| Finland | 0.23 | 0.27 | 0.37 | 0.70 | 0.33 | 0.27 | 0.08 | 0.26 | 0.36 | 0.41 | 0.41 | 0.35 |
| Sweden | 0.62 | 0.36 | 0.79 | 0.84 | 0.70 | 0.53 | 0.46 | 0.42 | 0.47 | 0.66 | 0.41 | 0.48 |
| Norway | 28.15 | 22.36 | 20.82 | 24.15 | 24.55 | 21.63 | 19.36 | 18.57 | 15.13 | 17.60 | 20.82 | 15.71 |

## Appendix B

**Table A2.** Correlation between the volume dynamics of mining and manufacturing industries (value added at factor cost, in million euro) for 2008–2019, $p < 0.05$. Source: own elaboration based on Eurostat (2022b).

|  | *R* | *t*-Test | **Reliability of Correlation** |
|---|---|---|---|
| Bulgaria | 0.194 | 4.1840 | No |
| Czechia | −0.6995 | 5.4976 | Yes |
| Denmark | −0.3 | 2.1134 | No |
| Belgium | −0.51 | 4.3922 | No |
| Germany | −0.702 | 7.2902 | Yes |
| Estonia | 0.747 | 1.2966 | Yes |
| Ireland | 0.0001 | 6.8833 | No |
| Greece | −0.12 | 2.2076 | No |
| Spain | 0.314 | 0.58654 | No |
| France | −0.84 | 7.0469 | Yes |
| Croatia | −0.90 | 7.7919 | Yes |
| Italy | −0.765 | 5.3466 | Yes |
| Cyprus | 0.498 | −0.6566 | No |
| Latvia | 0.937 | 0.04727 | Yes |
| Lithunania | 0.232 | 5.3818 | No |
| Luxemburg | 0.249 | 4.7387 | No |
| Hungary | 0.547 | 0.30675 | No |
| Netherlands | −0.874 | 8.3166 | Yes |
| Austria | −0.42 | 4.4649 | No |
| Poland | 0.007 | 5.0537 | No |
| Portugal | −0.19 | 3.9199 | No |
| Romania | −0.755 | 6.3407 | Yes |
| Slovenia | 0.170 | 4.8757 | No |
| Slovakia | −0.18 | 16.165 | No |
| Finland | 0.222 | 0.09862 | No |
| Sweden | 0.425 | −0.2088 | No |
| Norway | 0.715 | −1.393 | Yes |

**Table A3.** Correlation between the volume dynamics of mining industry and services in the total (value added at factor cost, in million euro) for 2008–2019, $p < 0.05$. Source: own elaboration based on Eurostat (2022b).

|  | *R* | *t*-Test | **Reliability of Correlation** |
|---|---|---|---|
| Bulgaria | 0.250 | 1.7458 | No |
| Czechia | −0.563 | 4.0303 | No |
| Denmark | −0.5062 | 2.5075 | No |
| Belgium | −0.5927 | 4.5700 | No |
| Germany | −0.826 | 7.6528 | Yes |
| Estonia | 0.613 | 1.3951 | Yes |
| Ireland | 0.023 | 3.2881 | No |
| Greece | −0.266 | 2.1904 | No |
| Spain | 0.142 | 0.4458 | No |
| France | −0.862 | 6.5121 | Yes |
| Croatia | −0.718 | 3.5611 | Yes |
| Italy | −0.888 | 6.8903 | Yes |
| Cyprus | −0.5798 | 2.9366 | Yes |
| Latvia | 0.7745 | −0.7139 | Yes |
| Lithunania | 0.518 | 1.6837 | No |
| Luxemburg | 0.110 | 5.8733 | No |
| Hungary | 0.732 | −1.157 | Yes |

**Table A3.** *Cont.*

|  | *R* | *t*-Test | **Reliability of Correlation** |
|---|---|---|---|
| Netherlands | −0.8798 | 7.6160 | Yes |
| Austria | −0.56 | 4.6735 | No |
| Poland | −0.02 | 3.4291 | No |
| Portugal | −0.09 | 2.2725 | No |
| Romania | −0.835 | 7.7867 | Yes |
| Slovenia | 0.470 | 1.6368 | No |
| Slovakia | −0.22 | 7.2030 | No |
| Finland | 0.206 | −0.1199 | No |
| Sweden | 0.174 | 0.8078 | No |
| Norway | 0.062 | 1.2585 | No |

**Appendix C**

**Table A4.** Dynamics of sector indicators specific for the production of goods and services for 2008–2019. Source: own elaboration based on Eurostat (2022a) ( . . . —no data).

|  | 2008 | 2009 | 2010 | 2011 | 2012 | 2013 | 2014 | 2015 | 2016 | 2017 | 2018 | 2019 |
|---|---|---|---|---|---|---|---|---|---|---|---|---|
| **Bulgaria** |  |  |  |  |  |  |  |  |  |  |  |  |
| million euro |  |  |  |  |  |  |  |  |  |  |  |  |
| Sector I | 469.5 | 462.7 | 616.5 | 810.3 | 848.7 | 664.2 | 588.9 | 590.7 | 581 | 726.1 | 687.9 | 652.9 |
| Sector II | 4278 | 3883.3 | 4123.5 | 4641.8 | 4677.9 | 4778.2 | 5315.2 | 6213.1 | 7135.1 | 7767.2 | 7819.7 | 8336.1 |
| Sector III | 6710.4 | 6082.6 | 6520.2 | 6795.7 | 7127.5 | 7721.2 | 7821.4 | 9044.7 | 9597.2 | 10,382.9 | 10,973.3 | 12,177.1 |
| Sector IV | 1479.6 | 1424.3 | 1407.6 | 1435.7 | 1624.4 | 1707.5 | 1747.5 | 2039.8 | 2313.6 | 2694.4 | 3075.4 | 3238.9 |
| Sector V | 5217.9 | 5006.8 | 5100.3 | 5304.1 | 5602.8 | 5808.3 | 6347.1 | 7372.5 | 8251 | 9433.2 | 10,909.7 | 13,070.6 |
| %% |  |  |  |  |  |  |  |  |  |  |  |  |
| Sector I | 2.59 | 2.74 | 3.47 | 4.27 | 4.27 | 3.21 | 2.70 | 2.34 | 2.08 | 2.34 | 2.06 | 1.74 |
| Sector II | 23.56 | 23.03 | 23.21 | 24.45 | 23.53 | 23.11 | 24.36 | 24.60 | 25.59 | 25.05 | 23.37 | 22.24 |
| Sector III | 36.96 | 36.08 | 36.70 | 35.79 | 35.85 | 37.34 | 35.84 | 35.81 | 34.43 | 33.49 | 32.79 | 32.49 |
| Sector IV | 8.15 | 8.45 | 7.92 | 7.56 | 8.17 | 8.26 | 8.01 | 8.07 | 8.30 | 8.69 | 9.19 | 8.64 |
| Sector V | 28.74 | 29.70 | 28.70 | 27.93 | 28.18 | 28.09 | 29.09 | 29.19 | 29.60 | 30.43 | 32.60 | 34.88 |
| **Czechia** |  |  |  |  |  |  |  |  |  |  |  |  |
| million euro |  |  |  |  |  |  |  |  |  |  |  |  |
| Sector I | 2181.8 | 1576.9 | 1713.5 | 1867.7 | 1602.4 | 1294.5 | 1148.6 | 1127.5 | 965.7 | 1120.2 | 1117.4 | 938.7 |
| Sector II | 31,734.1 | 26,175.3 | 29,002.4 | 31,611.4 | 31,463.7 | 31,457 | 33,590.4 | 3,5823.8 | 37,469.3 | 39,930.5 | 42,498.6 | 43,959 |
| Sector III | 27,588.7 | 24,623.9 | 26,405 | 27,122.5 | 25,915.4 | 25,984.9 | 24,850.6 | 27,115.5 | 28,009.1 | 30,993.6 | 34,202.6 | 35,554.9 |
| Sector IV | 7119.2 | 6565.8 | 6917.5 | 7165.6 | 7267.9 | 7376.8 | 7522 | 8114.4 | 8748.7 | 10,057.4 | 11,129.9 | 11,845.5 |
| Sector V | 27,747.1 | 24,878.2 | 25,325 | 26,779 | 26,413.4 | 25,879.9 | 25,946.1 | 28,169.1 | 29,669.8 | 33,376.4 | 37,065.4 | 40,444.2 |
| %% |  |  |  |  |  |  |  |  |  |  |  |  |
| Sector I | 2.26 | 1.88 | 1.92 | 1.98 | 1.73 | 1.41 | 1.23 | 1.12 | 0.92 | 0.97 | 0.89 | 0.71 |
| Sector II | 32.93 | 31.23 | 32.45 | 33.43 | 33.96 | 34.19 | 36.10 | 35.70 | 35.73 | 34.58 | 33.73 | 33.12 |
| Sector III | 28.63 | 29.38 | 29.55 | 28.69 | 27.97 | 28.25 | 26.70 | 27.02 | 26.71 | 26.84 | 27.14 | 26.78 |
| Sector IV | 7.39 | 7.83 | 7.74 | 7.58 | 7.84 | 8.02 | 8.08 | 8.09 | 8.34 | 8.71 | 8.83 | 8.92 |
| Sector V | 28.79 | 29.68 | 28.34 | 28.32 | 28.50 | 28.13 | 27.88 | 28.07 | 28.29 | 28.90 | 29.41 | 30.47 |
| **Denmark** |  |  |  |  |  |  |  |  |  |  |  |  |
| million euro |  |  |  |  |  |  |  |  |  |  |  |  |
| Sector I | 8541.4 | 5291.1 | 6214 | 7034.4 | 6319.5 | 5557.5 | 4271 | 4271.2 | 1906.3 | 2561.5 | 9922.5 | 2434.4 |
| Sector II | 28,681.5 | 24,846.9 | 26,197.1 | 27,203.3 | 28,764.5 | 29,364.8 | 29,796.7 | 32,518.2 | 33,722.4 | 35,623.4 | 38,106.9 | 41,212.1 |
| Sector III | 40,113.5 | 33,233.6 | 39,568.3 | 39,236.9 | 37,991.9 | 40,811.1 | 42,366.4 | 45,236 | 46,141 | 48,479.9 | 49,192.6 | 51,848 |
| Sector IV | 12,548.1 | 15,502.8 | 14,812.6 | 16,485.2 | 16,969.4 | 17,768.6 | 18,950.6 | 20,297.1 | 23,043 | 23,302 | 31,119 | 31,156.5 |
| Sector V | 48,974.8 | 52,329.2 | 52,655.5 | 53,468.6 | 54,930.5 | 57,176.3 | 60,719.7 | 64,292.4 | 69,269.8 | 72,826.9 | 80,540.5 | 84,122.6 |
| %% |  |  |  |  |  |  |  |  |  |  |  |  |
| Sector I | 6.15 | 4.03 | 4.46 | 4.90 | 4.36 | 3.69 | 2.74 | 2.56 | 1.10 | 1.40 | 4.75 | 1.15 |
| Sector II | 20.66 | 18.94 | 18.79 | 18.97 | 19.84 | 19.49 | 19.09 | 19.52 | 19.37 | 19.49 | 18.24 | 19.55 |
| Sector III | 28.89 | 25.33 | 28.38 | 27.36 | 26.21 | 27.08 | 27.14 | 27.15 | 26.51 | 26.52 | 23.55 | 24.60 |
| Sector IV | 9.04 | 11.82 | 10.62 | 11.49 | 11.70 | 11.79 | 12.14 | 12.18 | 13.24 | 12.75 | 14.90 | 14.78 |
| Sector V | 35.27 | 39.88 | 37.76 | 37.28 | 37.89 | 37.95 | 38.90 | 38.59 | 39.79 | 39.84 | 38.56 | 39.91 |

**Table A4.** *Cont.*

| | 2008 | 2009 | 2010 | 2011 | 2012 | 2013 | 2014 | 2015 | 2016 | 2017 | 2018 | 2019 |
|---|---|---|---|---|---|---|---|---|---|---|---|---|
| **Belgium** | | | | | | | | | | | | |
| million euro | | | | | | | | | | | | |
| Sector I | ... | ... | 290.2 | 310.8 | 271.4 | 259.2 | 292.1 | 235.8 | 257.5 | 271.2 | 188.3 | 269.8 |
| Sector II | 49,160.0 | 44,746.5 | 47,895.5 | 46,816.9 | 47,110.2 | 49,191.2 | 49,919.1 | 50,931.8 | 55,191.8 | 56,092.4 | 59,249.6 | 63,504.7 |
| Sector III | 51,737.5 | 55,157.4 | 63,516.6 | 65,637.5 | 67,593.5 | 65,675. | 65,407.3 | 66,201. | 68,075.5 | 72,960.6 | 73,583.1 | 83,561.4 |
| Sector IV | 18,938.5 | 20,732. | 22,029.1 | 24,161.9 | 25,023.2 | 25,356.4 | 26,799. | 30,204.1 | 32,160.4 | 33,344.3 | 33,340.6 | 37,027.7 |
| Sector V | ... | ... | ... | ... | ... | ... | ... | ... | ... | ... | ... | ... |
| %% | | | | | | | | | | | | |
| Sector I | ... | ... | ... | ... | ... | ... | ... | ... | ... | ... | ... | ... |
| Sector II | ... | ... | ... | ... | ... | ... | ... | ... | ... | ... | ... | ... |
| Sector III | ... | ... | ... | ... | ... | ... | ... | ... | ... | ... | ... | ... |
| Sector IV | ... | ... | ... | ... | ... | ... | ... | ... | ... | ... | ... | ... |
| Sector V | ... | ... | ... | ... | ... | ... | ... | ... | ... | ... | ... | ... |
| **Germany** | | | | | | | | | | | | |
| million euro | | | | | | | | | | | | |
| Sector I | 6766.1 | 5621.2 | 6834.7 | 5843.8 | 6157.2 | 5487.0 | 5240.1 | 5128.7 | 4470.5 | 4784.7 | 5522.0 | 3717.4 |
| Sector II | 453,779.2 | 381,547.6 | 455,788.2 | 490,206.5 | 481,846.9 | 490,616.9 | 519,792.5 | 534,931.9 | 569,863.5 | 592,027.2 | 650,200.9 | 658,751.4 |
| Sector III | 328,599.7 | 370,933.7 | 348 908.8 | 413,558.8 | 409,860.3 | 406,875.4 | 434,592.3 | 449,969.4 | 478,382.6 | 486,823. | 516,876.2 | 542,014.2 |
| Sector IV | 165,591 | 162,235.3 | 167,227.8 | 178,389 | 183,926.1 | 196,328.9 | 222,133.7 | 214,260 | 231,800.5 | 248,284.4 | 259,432 | 264,385.9 |
| Sector V | 517,348.8 | 491,871.9 | 512,695.7 | 552,349 | 566,140.8 | 621,568.6 | 675,617.6 | 666,506.1 | 706,678.9 | 756,367.2 | 768,260.9 | 785,734.9 |
| %% | | | | | | | | | | | | |
| Sector I | 0.46 | 0.40 | 0.46 | 0.36 | 0.37 | 0.32 | 0.28 | 0.27 | 0.22 | 0.23 | 0.25 | 0.16 |
| Sector II | 30.83 | 27.02 | 30.56 | 29.88 | 29.24 | 28.51 | 27.99 | 28.59 | 28.62 | 28.35 | 29.55 | 29.22 |
| Sector III | 22.32 | 26.27 | 23.39 | 25.21 | 24.87 | 23.64 | 23.40 | 24.05 | 24.02 | 23.31 | 23.49 | 24.04 |
| Sector IV | 11.25 | 11.49 | 11.21 | 10.88 | 11.16 | 11.41 | 11.96 | 11.45 | 11.64 | 11.89 | 11.79 | 11.73 |
| Sector V | 35.14 | 34.83 | 34.38 | 33.67 | 34.35 | 36.12 | 36.37 | 35.63 | 35.49 | 36.22 | 34.92 | 34.85 |
| **Estonia** | | | | | | | | | | | | |
| million euro | | | | | | | | | | | | |
| Sector I | 114.2 | 119.0 | 139.7 | 154.8 | 154.0 | 211.0 | 213.4 | 187.8 | 178.9 | 215.5 | 204.6 | 182.2 |
| Sector II | 2153.0 | 1582.0 | 1903.8 | 2296.8 | 2347.3 | 2476.9 | 2659.6 | 2693.6 | 2862.7 | 3018.2 | 3314.9 | 3423.8 |
| Sector III | 3028.6 | 2581. | 2777.5 | 3215.9 | 3517.6 | 3708.9 | 3918. | 3889.3 | 4212.1 | 4408.1 | 4834.8 | 4949.9 |
| Sector IV | 975.2 | 861.2 | 944.3 | 1260.5 | 1251.8 | 1393.2 | 1467.4 | 1587.7 | 1773.8 | 1974.6 | 1942.9 | 2286.6 |
| Sector V | ... | ... | ... | ... | ... | ... | ... | ... | ... | ... | ... | ... |
| %% | | | | | | | | | | | | |
| Sector I | ... | ... | ... | ... | ... | ... | ... | ... | ... | ... | ... | ... |
| Sector II | ... | ... | ... | ... | ... | ... | ... | ... | ... | ... | ... | ... |
| Sector III | ... | ... | ... | ... | ... | ... | ... | ... | ... | ... | ... | ... |
| Sector IV | ... | ... | ... | ... | ... | ... | ... | ... | ... | ... | ... | ... |
| Sector V | ... | ... | ... | ... | ... | ... | ... | ... | ... | ... | ... | ... |
| **Greece** | | | | | | | | | | | | |
| million euro | | | | | | | | | | | | |
| Sector I | 357.2 | 348.9 | 358.3 | 328.9 | 254.9 | 298.7 | 350.1 | 357.1 | 416.1 | 780.0 | 456.5 | 304.6 |
| Sector II | 17,041.1 | 16,901.2 | 15,873.2 | 13,629.1 | 11,873.5 | 10,288.3 | 9687.8 | 10,911.4 | 10,425.7 | 11,521.2 | 11,849.9 | 11,951.3 |
| Sector III | 33,163. | 33,405.8 | 31,008.8 | 27,139.3 | 23,180.9 | 21,780.6 | 19,121.9 | 19,925. | 20,827.6 | 22,059 | 20,346.7 | 20,644. |
| Sector IV | 7689.5 | 7826.4 | 7275.6 | 6624.4 | 5734.1 | 5281.7 | 6090.4 | 5949.4 | 5203.8 | 5984.7 | 6922.3 | 8333.2 |
| Sector V | 28,037.7 | 26,689.5 | 20,835.2 | 19,100.6 | 18,045.6 | 15,492.6 | 15,279.4 | 15,584.6 | 13,795.8 | 15,729.7 | 16,742.8 | 18,048.7 |
| %% | | | | | | | | | | | | |
| Sector I | 0.41 | 0.41 | 0.48 | 0.49 | 0.43 | 0.56 | 0.69 | 0.68 | 0.82 | 1.39 | 0.81 | 0.51 |
| Sector II | 19.75 | 19.84 | 21.07 | 20.40 | 20.09 | 19.36 | 19.17 | 20.69 | 20.58 | 20.55 | 21.04 | 20.16 |
| Sector III | 38.43 | 39.22 | 41.15 | 40.61 | 39.23 | 40.99 | 37.84 | 37.79 | 41.11 | 39.34 | 36.13 | 34.82 |
| Sector IV | 8.91 | 9.19 | 9.66 | 9.91 | 9.70 | 9.94 | 12.05 | 11.28 | 10.27 | 10.67 | 12.29 | 14.06 |
| Sector V | 32.49 | 31.34 | 27.65 | 28.58 | 30.54 | 29.15 | 30.24 | 29.56 | 27.23 | 28.05 | 29.73 | 30.45 |
| **Spain** | | | | | | | | | | | | |
| million euro | | | | | | | | | | | | |
| Sector I | 2472.9 | 1938.4 | 1953.0 | 1976.2 | 1847.8 | 1495.9 | 1365.6 | 1267.4 | 1290.4 | 2021.3 | 1634.2 | 1400.8 |
| Sector II | 126,704.3 | 100,824.6 | 106,153.4 | 103,869.8 | 95,650.5 | 93,133.7 | 97,577.3 | 101,928.0 | 105,309.8 | 110,841.4 | 120,875.6 | 125,438.6 |
| Sector III | 177,167.7 | 162,178.2 | 165,929.5 | 169,538.1 | 163,892.9 | 160,403. | 163,667. | 175,373.3 | 180,129.2 | 183,987.4 | 191,085.4 | 201,225.3 |
| Sector IV | 72,620.7 | 65,864.7 | 66,740.3 | 67246 | 65,654.4 | 64,931.6 | 66,867.9 | 73,400.3 | 76,698.3 | 84,277.1 | 88,063.8 | 95,668.3 |
| Sector V | 184,625.4 | 173,168.6 | 172,269.8 | 169,350.8 | 160,519.9 | 157,501.5 | 165,999.8 | 176,647.1 | 181,934.6 | 194,584.8 | 202,330.1 | 222,945.7 |
| %% | | | | | | | | | | | | |
| Sector I | 0.44 | 0.38 | 0.38 | 0.39 | 0.38 | 0.31 | 0.28 | 0.24 | 0.24 | 0.35 | 0.27 | 0.22 |
| Sector II | 22.48 | 20.01 | 20.69 | 20.29 | 19.62 | 19.51 | 19.69 | 19.28 | 19.31 | 19.25 | 20.01 | 19.40 |
| Sector III | 31.44 | 32.18 | 32.34 | 33.11 | 33.61 | 33.59 | 33.03 | 33.18 | 33.03 | 31.96 | 31.64 | 31.12 |
| Sector IV | 12.89 | 13.07 | 13.01 | 13.13 | 13.47 | 13.60 | 13.50 | 13.89 | 14.06 | 14.64 | 14.58 | 14.79 |
| Sector V | 32.76 | 34.36 | 33.58 | 33.08 | 32.92 | 32.99 | 33.50 | 33.42 | 33.36 | 33.80 | 33.50 | 34.48 |

**Table A4.** *Cont.*

| | 2008 | 2009 | 2010 | 2011 | 2012 | 2013 | 2014 | 2015 | 2016 | 2017 | 2018 | 2019 |
|---|---|---|---|---|---|---|---|---|---|---|---|---|
| **France** | | | | | | | | | | | | |
| million euro | | | | | | | | | | | | |
| Sector I | 2752.4 | 2353.4 | 2227.4 | 2582.1 | 2316.3 | 2175.0 | 1834.0 | 1813.0 | 1464.4 | 1151.2 | 1150.4 | 1335.7 |
| Sector II | 203,255.5 | 180,452.0 | 193,928.7 | 195,284.6 | 193,437.4 | 192,888.7 | 201,020.1 | 208,141.9 | 213,731.6 | 240,268.1 | 241,204.6 | 251,010.4 |
| Sector III | 275,954.8 | 266,204.6 | 281,369.9 | 287,851.3 | 289,888.6 | 291,231.4 | 287,636.2 | 299,107.9 | 306,551. | 313,152.7 | 318,555.9 | 331,008.4 |
| Sector IV | 132,628.7 | 127,072.4 | 139,023. | 146,687.7 | 143,012.6 | 145,108.4 | 143,396.5 | 150,919.5 | 157,936.9 | 157,937.5 | 163,256.2 | 166,201.6 |
| Sector V | 398,671.5 | 402,745.1 | 416,788.8 | 430,394.3 | 428,364. | 426,675. | 414,075.7 | 437,299.8 | 450,673.1 | 450,491. | 459,613.1 | 477,920.9 |
| %% | | | | | | | | | | | | |
| Sector I | 0.27 | 0.24 | 0.22 | 0.24 | 0.22 | 0.21 | 0.18 | 0.17 | 0.13 | 0.10 | 0.10 | 0.11 |
| Sector II | 20.06 | 18.44 | 18.77 | 18.37 | 18.30 | 18.23 | 19.18 | 18.97 | 18.91 | 20.66 | 20.38 | 20.45 |
| Sector III | 27.23 | 27.20 | 27.23 | 27.08 | 27.43 | 27.52 | 27.45 | 27.26 | 27.12 | 26.93 | 26.91 | 26.97 |
| Sector IV | 13.09 | 12.98 | 13.45 | 13.80 | 13.53 | 13.71 | 13.68 | 13.75 | 13.97 | 13.58 | 13.79 | 13.54 |
| Sector V | 39.35 | 41.15 | 40.33 | 40.50 | 40.53 | 40.33 | 39.51 | 39.85 | 39.87 | 38.74 | 38.83 | 38.94 |
| **Croatia** | | | | | | | | | | | | |
| million euro | | | | | | | | | | | | |
| Sector I | . . . | 944.4 | 1288.6 | 1547.4 | 1328.7 | 1223.7 | 1147.6 | 819.9 | 137.6 | 141.9 | 165.5 | 74.0 |
| Sector II | 6103.9 | 5164.1 | 4951.1 | 4786.9 | 4585.4 | 4541.6 | 4646.5 | 4924.3 | 5767.6 | 6126.4 | 6249.3 | 7007.2 |
| Sector III | 8545.7 | 7514.4 | 7379. | 7049.8 | 6636. | 6950.7 | 7664. | 8301.8 | 8500.5 | 8483.2 | 9382.2 | 10,331.2 |
| Sector IV | 1969.7 | 1950.3 | 1937.7 | 1984.2 | 1977.9 | 2349.2 | 2392.7 | 2515.3 | 3039.1 | 3170.6 | 3616. | 3712.8 |
| Sector V | . . . | . . . | . . . | . . . | . . . | . . . | . . . | . . . | . . . | . . . | . . . | . . . |
| %% | | | | | | | | | | | | |
| Sector I | . . . | . . . | . . . | . . . | . . . | . . . | . . . | . . . | . . . | . . . | . . . | . . . |
| Sector II | . . . | . . . | . . . | . . . | . . . | . . . | . . . | . . . | . . . | . . . | . . . | . . . |
| Sector III | . . . | . . . | . . . | . . . | . . . | . . . | . . . | . . . | . . . | . . . | . . . | . . . |
| Sector IV | . . . | . . . | . . . | . . . | . . . | . . . | . . . | . . . | . . . | . . . | . . . | . . . |
| Sector V | . . . | . . . | . . . | . . . | . . . | . . . | . . . | . . . | . . . | . . . | . . . | . . . |
| **Italy** | | | | | | | | | | | | |
| million euro | | | | | | | | | | | | |
| Sector I | 5864.0 | 5174.3 | 5682.9 | 5045.5 | 4706.8 | 3941.3 | 3715.8 | 3591.3 | 3407.9 | 2357.5 | 2715.2 | 2430.4 |
| Sector II | 211,744.7 | 180,256.8 | 205,589.3 | 208,093.5 | 199,296.5 | 198,678.9 | 204,053.7 | 212,949.5 | 224,994.7 | 241,413.9 | 246,941.3 | 250,193.2 |
| Sector III | 194,358.3 | 185,785.9 | 213,569. | 219,949.6 | 210,648.8 | 210,165.7 | 213,687.1 | 223,046.9 | 233,657.4 | 237,380.4 | 244,162.1 | 255,028.1 |
| Sector IV | 72,081.5 | 67,938.1 | 76,407.7 | 80,061.7 | 78,907.2 | 77,086.2 | 78,330. | 82,961.4 | 87,768. | 91,752.2 | 97,843.8 | 99,627.7 |
| Sector V | 243,806.3 | 212,968.7 | 245,701.1 | 241,006.3 | 232,498.1 | 227,471.1 | 230,339. | 239,710.5 | 249,880.7 | 252,887.8 | 268,216. | 269,443.7 |
| %% | | | | | | | | | | | | |
| Sector I | 0.81 | 0.79 | 0.76 | 0.67 | 0.65 | 0.55 | 0.51 | 0.47 | 0.43 | 0.29 | 0.32 | 0.28 |
| Sector II | 29.09 | 27.64 | 27.52 | 27.59 | 27.45 | 27.70 | 27.95 | 27.94 | 28.13 | 29.23 | 28.72 | 28.54 |
| Sector III | 26.70 | 28.49 | 28.59 | 29.16 | 29.01 | 29.30 | 29.27 | 29.26 | 29.22 | 28.75 | 28.39 | 29.09 |
| Sector IV | 9.90 | 10.42 | 10.23 | 10.62 | 10.87 | 10.75 | 10.73 | 10.88 | 10.97 | 11.11 | 11.38 | 11.36 |
| Sector V | 33.50 | 32.66 | 32.89 | 31.96 | 32.02 | 31.71 | 31.55 | 31.45 | 31.25 | 30.62 | 31.19 | 30.73 |
| **Cyprus** | | | | | | | | | | | | |
| million euro | | | | | | | | | | | | |
| Sector I | 59.4 | 45.4 | 44.8 | 34.6 | 16.4 | 18.8 | 21.9 | 26.1 | 16.2 | 14.9 | 21.4 | 22.9 |
| Sector II | 1228.1 | 1188.6 | 1158.3 | 1057.1 | 951.2 | 818.9 | 823.7 | 887.4 | 962.8 | 1077.6 | 1201.8 | 1318.9 |
| Sector III | 3270.8 | 3211.2 | 3281.8 | 3138. | 2986.4 | 2749.2 | 2695.6 | 2753.5 | 2918.2 | 3084.6 | 3231.2 | 3448. |
| Sector IV | 1305.5 | 1226.3 | 1247. | 1296.2 | 1322.7 | 1230.9 | 1267.9 | 1317.8 | 1469.6 | 1591.5 | 1721.5 | 1886.5 |
| Sector V | 2934.7 | 2978.6 | 3173.2 | 3180.7 | 3114.3 | 3141.3 | 3339.6 | 3461.8 | 3939.6 | 4369.9 | 4780.7 | 5609.1 |
| %% | | | | | | | | | | | | |
| Sector I | 0.68 | 0.52 | 0.50 | 0.40 | 0.20 | 0.24 | 0.27 | 0.31 | 0.17 | 0.15 | 0.20 | 0.19 |
| Sector II | 13.96 | 13.74 | 13.01 | 12.14 | 11.34 | 10.29 | 10.11 | 10.51 | 10.35 | 10.63 | 10.97 | 10.74 |
| Sector III | 37.17 | 37.12 | 36.85 | 36.04 | 35.59 | 34.54 | 33.08 | 32.60 | 31.36 | 30.42 | 29.49 | 28.07 |
| Sector IV | 14.84 | 14.18 | 14.00 | 14.89 | 15.76 | 15.47 | 15.56 | 15.60 | 15.79 | 15.70 | 15.71 | 15.36 |
| Sector V | 33.35 | 34.43 | 35.63 | 36.53 | 37.11 | 39.47 | 40.98 | 40.98 | 42.33 | 43.10 | 43.63 | 45.66 |
| **Latvia** | | | | | | | | | | | | |
| million euro | | | | | | | | | | | | |
| Sector I | 59.3 | 58.2 | 59.3 | 68.8 | 78.8 | 80.5 | 77.2 | 85.3 | 86.6 | 102.0 | 111.8 | 112.6 |
| Sector II | 1875.5 | 1230.5 | 1540.2 | 1626.6 | 1877.8 | 1883.1 | 1958.5 | 2073.5 | 2156.3 | 2333.9 | 2589.1 | 2823.5 |
| Sector III | 4732.1 | 3516. | 3672.9 | 3860.5 | 4392.2 | 4438.9 | 4596.4 | 4816.2 | 5132.5 | 5196.8 | 5558.6 | 5881.7 |
| Sector IV | 1303.3 | 887.2 | 894.1 | 966.4 | 1054.7 | 1189.5 | 1281.6 | 1352. | 1440.5 | 1439. | 1755.2 | 1886.3 |
| Sector V | 3412.2 | 2554.6 | 2314.8 | 2381.9 | 2855.3 | 3137.8 | 3435.5 | 3624.3 | 3993.9 | 4190.7 | 4978.2 | 5594.9 |
| %% | | | | | | | | | | | | |
| Sector I | 0.52 | 0.71 | 0.70 | 0.77 | 0.77 | 0.75 | 0.68 | 0.71 | 0.68 | 0.77 | 0.75 | 0.69 |
| Sector II | 16.48 | 14.92 | 18.16 | 18.27 | 18.30 | 17.55 | 17.26 | 17.35 | 16.83 | 17.60 | 17.27 | 17.32 |
| Sector III | 41.57 | 42.64 | 43.31 | 43.36 | 42.81 | 41.37 | 40.50 | 40.30 | 40.07 | 39.18 | 37.07 | 36.09 |
| Sector IV | 11.45 | 10.76 | 10.54 | 10.85 | 10.28 | 11.09 | 11.29 | 11.31 | 11.25 | 10.85 | 11.71 | 11.57 |
| Sector V | 29.98 | 30.98 | 27.29 | 26.75 | 27.83 | 29.24 | 30.27 | 30.33 | 31.18 | 31.60 | 33.20 | 34.33 |

**Table A4.** *Cont.*

| | 2008 | 2009 | 2010 | 2011 | 2012 | 2013 | 2014 | 2015 | 2016 | 2017 | 2018 | 2019 |
|---|---|---|---|---|---|---|---|---|---|---|---|---|
| **Lithunania** | | | | | | | | | | | | |
| million euro | | | | | | | | | | | | |
| Sector I | 112.8 | 62.3 | 71.2 | 100.7 | 91.0 | 94.2 | 89.3 | 75.3 | 79.8 | 90.6 | 101.2 | 100.9 |
| Sector II | 2709.1 | 2178.4 | 2507.4 | 2905.8 | 2878.8 | 2875.3 | 3280.0 | 4034.5 | 5213.0 | 4552.7 | 4984.7 | 5532.4 |
| Sector III | 5607.2 | 3875.4 | 4231.3 | 5323.6 | 5700.9 | 5865.9 | 6790.4 | 6777.9 | 7285.1 | 8043.9 | 9123.3 | 9939.7 |
| Sector IV | 1165.5 | 923.9 | 837.4 | 1028.2 | 1143.2 | 1192. | 1440.5 | 1669.5 | 1915.5 | 2176.4 | 2565.6 | 2613.1 |
| Sector V | 3403.9 | 2654.7 | 2622.3 | 3183.3 | 3410.8 | 3545.7 | 4170.9 | 4726.1 | 5185.8 | 5821.8 | 6551.4 | 7145.1 |
| %% | | | | | | | | | | | | |
| Sector I | 0.87 | 0.64 | 0.69 | 0.80 | 0.69 | 0.69 | 0.57 | 0.44 | 0.41 | 0.44 | 0.43 | 0.40 |
| Sector II | 20.84 | 22.47 | 24.42 | 23.17 | 21.77 | 21.18 | 20.80 | 23.34 | 26.49 | 22.01 | 21.37 | 21.84 |
| Sector III | 43.14 | 39.97 | 41.20 | 42.45 | 43.11 | 43.22 | 43.06 | 39.22 | 37.02 | 38.89 | 39.11 | 39.24 |
| Sector IV | 8.97 | 9.53 | 8.15 | 8.20 | 8.64 | 8.78 | 9.13 | 9.66 | 9.73 | 10.52 | 11.00 | 10.32 |
| Sector V | 26.19 | 27.38 | 25.53 | 25.38 | 25.79 | 26.12 | 26.45 | 27.34 | 26.35 | 28.14 | 28.09 | 28.21 |
| **Luxemburg** | | | | | | | | | | | | |
| million euro | | | | | | | | | | | | |
| Sector I | 32.2 | 31.7 | 32.7 | 36.2 | 32.0 | 31.5 | 29.9 | 27.0 | 30.6 | 33.0 | 33.7 | 37.5 |
| Sector II | 2982.8 | 1988.0 | 2383.4 | 2487.4 | 2409.0 | 2375.3 | 2523.5 | 2600.3 | 3118.2 | 3125.2 | 3170.5 | 3072.5 |
| Sector III | 5428.6 | 4800.4 | 5767.1 | 6647.9 | 6435.9 | 6606.6 | 7419.3 | 6899.7 | 7430.4 | 7799.6 | 8184.3 | 7997.8 |
| Sector IV | 2857.8 | 2946.4 | 2682.8 | 2893.1 | 3099.5 | 3131.7 | 3373.9 | 3567.3 | 3703. | 4069.5 | 4429.3 | 4767.1 |
| Sector V | … | … | … | … | … | … | … | … | … | … | … | … |
| %% | | | | | | | | | | | | |
| Sector I | … | … | … | … | … | … | … | … | … | … | … | … |
| Sector II | … | … | … | … | … | … | … | … | … | … | … | … |
| Sector III | … | … | … | … | … | … | … | … | … | … | … | … |
| Sector IV | … | … | … | … | … | … | … | … | … | … | … | … |
| Sector V | … | … | … | … | … | … | … | … | … | … | … | … |
| **Netherlands** | | | | | | | | | | | | |
| million euro | | | | | | | | | | | | |
| Sector I | 8886.7 | 9670.3 | 9728.8 | 10,090.2 | 11,324.8 | 11,048.2 | 9269.6 | 8137.8 | 5272.5 | 5301.5 | 5676.5 | 4857.3 |
| Sector II | 59,313.6 | 52,935.7 | 58,360.8 | 60,600.1 | 59,907.1 | 57,776.9 | 58,676.8 | 63,067.1 | 67,208.2 | 70,850.4 | 74,284.4 | 77,518.1 |
| Sector III | 100,817.6 | 97,720.5 | 103,746.8 | 107,752.9 | 106,759.4 | 110,282. | 110,159. | 115,634.6 | 119,191.6 | 124,869.7 | 131,672.4 | 142,894.5 |
| Sector IV | 40,166.1 | 41,960. | 41,618.3 | 43,732.8 | 46,287.9 | 47,615.5 | 48,887.3 | 54,045.4 | 57,963.3 | 62,344.7 | 69,429.7 | 75,610.3 |
| Sector V | … | … | … | … | … | … | … | … | … | … | … | … |
| %% | | | | | | | | | | | | |
| Sector I | … | … | … | … | … | … | … | … | … | … | … | … |
| Sector II | … | … | … | … | … | … | … | … | … | … | … | … |
| Sector III | … | … | … | … | … | … | … | … | … | … | … | … |
| Sector IV | … | … | … | … | … | … | … | … | … | … | … | … |
| Sector V | … | … | … | … | … | … | … | … | … | … | … | … |
| **Hungary** | | | | | | | | | | | | |
| million euro | | | | | | | | | | | | |
| Sector I | 213.6 | 200.7 | 151.6 | 189.7 | 183.1 | 201.0 | 171.0 | 147.5 | 136.9 | 193.1 | 259.2 | 327.4 |
| Sector II | 19,278.5 | 15,447.7 | 17,495.5 | 18,817.1 | 18,019.8 | 18,585.0 | 19,881.0 | 21,918.8 | 21,785.1 | 24,176.0 | 26,097.3 | 26,232.0 |
| Sector III | 16,450. | 13,670.8 | 14,155.7 | 14,763.6 | 14,250.7 | 14,985.3 | 15,658.5 | 16,926.6 | 17,723.3 | 18,982.4 | 20,987.9 | 22,132.1 |
| Sector IV | 4926.2 | 4286.5 | 4746.7 | 4961.1 | 4672.2 | 4736.4 | 5125.2 | 5657.2 | 6145.4 | 7107.2 | 8314.5 | 8646.6 |
| Sector V | 15,990.5 | 14,464.1 | 16,098.7 | 16,653.4 | 16,353. | 17,090.4 | 17,886.5 | 19,134. | 20,144.4 | 23,595.8 | 26,028. | 27,797.7 |
| %% | | | | | | | | | | | | |
| Sector I | 0.38 | 0.42 | 0.29 | 0.34 | 0.34 | 0.36 | 0.29 | 0.23 | 0.21 | 0.26 | 0.32 | 0.38 |
| Sector II | 33.91 | 32.14 | 33.23 | 33.98 | 33.70 | 33.43 | 33.86 | 34.36 | 33.04 | 32.65 | 31.95 | 30.81 |
| Sector III | 28.93 | 28.44 | 26.89 | 26.66 | 26.65 | 26.95 | 26.67 | 26.54 | 26.88 | 25.63 | 25.69 | 26.00 |
| Sector IV | 8.66 | 8.92 | 9.02 | 8.96 | 8.74 | 8.52 | 8.73 | 8.87 | 9.32 | 9.60 | 10.18 | 10.16 |
| Sector V | 28.12 | 30.09 | 30.58 | 30.07 | 30.58 | 30.74 | 30.46 | 30.00 | 30.55 | 31.86 | 31.86 | 32.65 |
| **Austria** | | | | | | | | | | | | |
| million euro | | | | | | | | | | | | |
| Sector I | 1150.8 | 950.3 | 1026.4 | 1295.2 | 1294.0 | 1245.4 | 1162.4 | 991.5 | 838.2 | 952.2 | 1022.6 | 918.1 |
| Sector II | 46,702.3 | 41,218.4 | 45,139.5 | 48,392.1 | 48,315.3 | 47,493.2 | 49,257.3 | 51,585.0 | 54,390.3 | 56,265.9 | 61,584.6 | 62,336.3 |
| Sector III | 48,424.4 | 46,367.9 | 48,753. | 52,065.4 | 52,796.1 | 54,290.1 | 53,171.4 | 55,260.5 | 57,646.2 | 59,786. | 62,278.4 | 63,874.5 |
| Sector IV | 23,028.7 | 22,276.8 | 23,405.9 | 24,990.9 | 26,380.8 | 27,327. | 27,362.8 | 28,258.8 | 30,671.7 | 32,775.3 | 31,925.4 | 33,035.7 |
| Sector V | 51,648.2 | 50,022.1 | 51,690.6 | 55,362.5 | 57,933.1 | 61,344. | 62,047.9 | 64,913. | 68,851.9 | 71,720.7 | 67,815.7 | 71,493.1 |
| %% | | | | | | | | | | | | |
| Sector I | 0.67 | 0.59 | 0.60 | 0.71 | 0.69 | 0.65 | 0.60 | 0.49 | 0.39 | 0.43 | 0.46 | 0.40 |
| Sector II | 27.32 | 25.63 | 26.55 | 26.57 | 25.88 | 24.77 | 25.52 | 25.66 | 25.61 | 25.40 | 27.42 | 26.91 |
| Sector III | 28.33 | 28.83 | 28.68 | 28.59 | 28.28 | 28.32 | 27.55 | 27.49 | 27.14 | 26.99 | 27.73 | 27.57 |
| Sector IV | 13.47 | 13.85 | 13.77 | 13.72 | 14.13 | 14.26 | 14.18 | 14.06 | 14.44 | 14.80 | 14.21 | 14.26 |
| Sector V | 30.21 | 31.10 | 30.40 | 30.40 | 31.03 | 32.00 | 32.15 | 32.29 | 32.42 | 32.38 | 30.19 | 30.86 |

**Table A4.** *Cont.*

| | 2008 | 2009 | 2010 | 2011 | 2012 | 2013 | 2014 | 2015 | 2016 | 2017 | 2018 | 2019 |
|---|---|---|---|---|---|---|---|---|---|---|---|---|
| **Poland** | | | | | | | | | | | | |
| million euro | | | | | | | | | | | | |
| Sector I | 8735.8 | 6564.5 | 7866.7 | 9549.6 | 9007.5 | 7813.3 | 7329.5 | 7499.4 | 6520.4 | 9106.3 | 8113.1 | 7550.6 |
| Sector II | 57,212.3 | 45,725.8 | 49,480.3 | 54,113.6 | 52,422.7 | 54,564.3 | 57,725.3 | 61,897.4 | 64,219.7 | 70,361.0 | 80,073.3 | 85,821.0 |
| Sector III | 66,523.2 | 53,587.2 | 61,988.6 | 62,006.2 | 62,085.3 | 61,203.4 | 65,460. | 67,458.3 | 70,289.9 | 76,031.6 | 85,501.7 | 93,846.4 |
| Sector IV | 12,091.5 | 10,499.8 | 12,283.4 | 12,860.8 | 13,064.1 | 11,988.1 | 14,067.6 | 15,063.9 | 15,540.2 | 17,668.2 | 25,145.5 | 26,622.9 |
| Sector V | ... | ... | ... | ... | ... | ... | ... | ... | ... | ... | ... | ... |
| %% | | | | | | | | | | | | |
| Sector I | ... | ... | ... | ... | ... | ... | ... | ... | ... | ... | ... | ... |
| Sector II | ... | ... | ... | ... | ... | ... | ... | ... | ... | ... | ... | ... |
| Sector III | ... | ... | ... | ... | ... | ... | ... | ... | ... | ... | ... | ... |
| Sector IV | ... | ... | ... | ... | ... | ... | ... | ... | ... | ... | ... | ... |
| Sector V | ... | ... | ... | ... | ... | ... | ... | ... | ... | ... | ... | ... |
| **Portugal** | | | | | | | | | | | | |
| million euro | | | | | | | | | | | | |
| Sector I | 510.3 | 522.7 | 559.1 | 522.5 | 462.0 | 418.8 | 414.1 | 388.5 | 395.0 | 453.4 | 469.5 | 461.3 |
| Sector II | 18,730.9 | 16,752.9 | 18,017.2 | 17,193.2 | 16,254.5 | 16,684.1 | 17,425.2 | 19,227.0 | 20,136.0 | 21,842.4 | 22,453.1 | 22,530.2 |
| Sector III | 28,974.9 | 28,569.1 | 28,632. | 26,673.5 | 25,248.9 | 25,625. | 26,723.1 | 27,459.3 | 28,866.9 | 29,946.1 | 31,676.7 | 33,339.9 |
| Sector IV | 11,625.7 | 11,202.8 | 11,009.8 | 10,387.3 | 8988.5 | 8979. | 9663.9 | 10,623.4 | 12,109.4 | 14,234.1 | 15,549.5 | 16,895. |
| Sector V | 26,772.9 | 26,532.5 | 26,546.1 | 24,998.3 | 23,296.3 | 23,186.7 | 24,387.2 | 24,985.7 | 26,808.1 | 29,576.6 | 32,426.7 | 35,259.4 |
| %% | | | | | | | | | | | | |
| Sector I | 0.59 | 0.63 | 0.66 | 0.65 | 0.62 | 0.56 | 0.53 | 0.47 | 0.45 | 0.47 | 0.46 | 0.43 |
| Sector II | 21.63 | 20.04 | 21.26 | 21.55 | 21.89 | 22.28 | 22.17 | 23.25 | 22.80 | 22.74 | 21.89 | 20.77 |
| Sector III | 33.45 | 34.18 | 33.78 | 33.44 | 34.01 | 34.22 | 33.99 | 33.21 | 32.69 | 31.18 | 30.88 | 30.73 |
| Sector IV | 13.42 | 13.40 | 12.99 | 13.02 | 12.11 | 11.99 | 12.29 | 12.85 | 13.71 | 14.82 | 15.16 | 15.57 |
| Sector V | 30.91 | 31.75 | 31.32 | 31.34 | 31.38 | 30.96 | 31.02 | 30.22 | 30.35 | 30.79 | 31.61 | 32.50 |
| **Romania** | | | | | | | | | | | | |
| million euro | | | | | | | | | | | | |
| Sector I | 3984.2 | 2480.2 | 2938.0 | 3478.2 | 3603.4 | 3604.1 | 3376.1 | 2589.0 | 1832.9 | 1425.0 | 1023.4 | 1254.5 |
| Sector II | 15,493.1 | 11,454.9 | 12,778.0 | 13,326.8 | 13,436.2 | 13,962.6 | 15,862.9 | 15,357.5 | 16,943.8 | 18,742.1 | 22,726.3 | 23,453.1 |
| Sector III | 20,333.4 | 16,223.6 | 16,828.4 | 17,238.6 | 17,343.3 | 19,077.8 | 20,001.7 | 20,394.4 | 23,450.2 | 24,593.2 | 26,897.8 | 29,324.3 |
| Sector IV | 4401.5 | 3313.2 | 3313. | 3632.7 | 3758.2 | 4035.9 | 4727.8 | 4985.4 | 5508.1 | 4917.2 | 7593.4 | 9137.7 |
| Sector V | 15 291. | 12,169.4 | 12,193.4 | 12,751.5 | 13,257.8 | 14,690.7 | 16,315.2 | 17,217.5 | 19,236.4 | 22,071.4 | 24,853.3 | 26,959.7 |
| %% | | | | | | | | | | | | |
| Sector I | 6.70 | 5.43 | 6.11 | 6.90 | 7.01 | 6.51 | 5.60 | 4.28 | 2.74 | 1.99 | 1.23 | 1.39 |
| Sector II | 26.04 | 25.10 | 26.59 | 26.43 | 26.14 | 25.22 | 26.31 | 25.37 | 25.30 | 26.12 | 27.35 | 26.02 |
| Sector III | 34.17 | 35.55 | 35.02 | 34.18 | 33.74 | 34.45 | 33.18 | 33.69 | 35.02 | 34.28 | 32.37 | 32.54 |
| Sector IV | 7.40 | 7.26 | 6.89 | 7.20 | 7.31 | 7.29 | 7.84 | 8.23 | 8.22 | 6.85 | 9.14 | 10.14 |
| Sector V | 25.70 | 26.66 | 25.38 | 25.29 | 25.79 | 26.53 | 27.06 | 28.44 | 28.72 | 30.76 | 29.91 | 29.91 |
| **Slovenia** | | | | | | | | | | | | |
| million euro | | | | | | | | | | | | |
| Sector I | 134.9 | 129.2 | 142.0 | 120.8 | 109.4 | 104.2 | 108.8 | 110.6 | 117.6 | 127.5 | 127.6 | 131.0 |
| Sector II | 6743.5 | 5320.7 | 6188.9 | 6326.8 | 6164.9 | 6290.7 | 6888.2 | 7159.7 | 7651.9 | 8365.7 | 8811.0 | 9165.7 |
| Sector III | 6469. | 5361.5 | 6142.9 | 6348.8 | 5917.5 | 5978.8 | 6123.6 | 6503.4 | 7016.8 | 7481.6 | 7896.7 | 8376. |
| Sector IV | 1363.5 | 1254.1 | 1288.8 | 1254.9 | 1226.2 | 1257.7 | 1418.8 | 1511.1 | 1700.7 | 1911. | 2128.2 | 2222.7 |
| Sector V | 5323.9 | 4827.8 | 4975.7 | 5009.1 | 4911.4 | 5089.7 | 5479. | 5740.5 | 6095.9 | 6831. | 7429. | 7992.1 |
| %% | | | | | | | | | | | | |
| Sector I | 0.67 | 0.76 | 0.76 | 0.63 | 0.60 | 0.56 | 0.54 | 0.53 | 0.52 | 0.52 | 0.48 | 0.47 |
| Sector II | 33.66 | 31.50 | 33.03 | 33.19 | 33.63 | 33.60 | 34.41 | 34.05 | 33.88 | 33.85 | 33.38 | 32.87 |
| Sector III | 32.29 | 31.74 | 32.78 | 33.31 | 32.28 | 31.94 | 30.59 | 30.93 | 31.07 | 30.27 | 29.92 | 30.03 |
| Sector IV | 6.81 | 7.42 | 6.88 | 6.58 | 6.69 | 6.72 | 7.09 | 7.19 | 7.53 | 7.73 | 8.06 | 7.97 |
| Sector V | 26.57 | 28.58 | 26.55 | 26.28 | 26.80 | 27.19 | 27.37 | 27.30 | 26.99 | 27.64 | 28.15 | 28.66 |
| **Slovakia** | | | | | | | | | | | | |
| million euro | | | | | | | | | | | | |
| Sector I | 310.3 | 316.9 | 299.3 | 324.8 | 314.3 | 318.1 | 318.3 | 345.7 | 290.1 | 298.2 | 316.4 | 296.7 |
| Sector II | 7969.3 | 6279.1 | 9696.6 | 10,076.1 | 9862.2 | 10,037.6 | 11,327.2 | 12,758.6 | 12,901.4 | 14,075.1 | 14,856.2 | 14,473.1 |
| Sector III | 9828. | 8681.7 | 11,954.2 | 13,181.9 | 12,749. | 10,449.6 | 10,987.4 | 11,180.9 | 11,405.7 | 11,778.9 | 12,431.1 | 13,442.3 |
| Sector IV | 1575.2 | 1644.5 | 2824.9 | 2559.2 | 2790.8 | 4093.3 | 2183.7 | 2540.1 | 3263.3 | 3476.5 | 3744.3 | 4034.2 |
| Sector V | ... | ... | ... | ... | ... | ... | ... | ... | ... | ... | ... | ... |
| %% | | | | | | | | | | | | |
| Sector I | ... | ... | ... | ... | ... | ... | ... | ... | ... | ... | ... | ... |
| Sector II | ... | ... | ... | ... | ... | ... | ... | ... | ... | ... | ... | ... |
| Sector III | ... | ... | ... | ... | ... | ... | ... | ... | ... | ... | ... | ... |
| Sector IV | ... | ... | ... | ... | ... | ... | ... | ... | ... | ... | ... | ... |
| Sector V | ... | ... | ... | ... | ... | ... | ... | ... | ... | ... | ... | ... |

**Table A4.** *Cont.*

| | 2008 | 2009 | 2010 | 2011 | 2012 | 2013 | 2014 | 2015 | 2016 | 2017 | 2018 | 2019 |
|---|---|---|---|---|---|---|---|---|---|---|---|---|
| **Finland** | | | | | | | | | | | | |
| million euro | | | | | | | | | | | | |
| Sector I | 397.7 | 424.6 | 602.0 | 1200.1 | 580.3 | 466.7 | 151.1 | 481.5 | 671.7 | 806.2 | 826.6 | 735.8 |
| Sector II | 32,089.1 | 22,713.7 | 26,505.1 | 25,797.9 | 23,760.8 | 24,507.5 | 24,610.7 | 25,154.4 | 23,983.5 | 29,753.2 | 29,385.4 | 28,627.0 |
| Sector III | 26,938.2 | 25,058.1 | 26,079. | 27,092.6 | 27,099.4 | 27,234.1 | 27,251.9 | 27,330.6 | 28,190. | 28,762.2 | 29,906.1 | 30,525.3 |
| Sector IV | 9957.5 | 9838.3 | 10,236.7 | 6821.8 | 11,661.3 | 11,001.4 | 11,237.2 | 11,799. | 12,808.2 | 13,441.9 | 14,565.4 | 15,177.2 |
| Sector V | … | … | … | … | … | … | … | … | … | … | … | … |
| %% | | | | | | | | | | | | |
| Sector I | … | … | … | … | … | … | … | … | … | … | … | … |
| Sector II | … | … | … | … | … | … | … | … | … | … | … | … |
| Sector III | … | … | … | … | … | … | … | … | … | … | … | … |
| Sector IV | … | … | … | … | … | … | … | … | … | … | … | … |
| Sector V | … | … | … | … | … | … | … | … | … | … | … | … |
| **Sweden** | | | | | | | | | | | | |
| million euro | | | | | | | | | | | | |
| Sector I | 1944.6 | 1003.9 | 2615.1 | 3056.5 | 2686.1 | 2,093.9 | 1779.9 | 1694.8 | 1927.7 | 2798.5 | 1716.8 | 2031.6 |
| Sector II | 50,261.2 | 39,112.9 | 50,798.6 | 53,970.0 | 53,204.8 | 52,660.2 | 51,824.4 | 52,988.0 | 53,528.4 | 55,108.7 | 59,971.2 | 59,697.7 |
| Sector III | 52,478.8 | 48,122.6 | 57,914.5 | 61,974.5 | 63,928.5 | 64,080.6 | 64,412.1 | 64,617.4 | 66,227.9 | 67,673.9 | 66,403.9 | 69,558.7 |
| Sector IV | 25,804.1 | 23,419.7 | 26,863.5 | 31,640.8 | 34,269.3 | 35,837.9 | 36,147.5 | 37,690.4 | 39,291.3 | 40,994.1 | 38,840.8 | 38,535. |
| Sector V | … | … | … | … | … | … | … | … | … | … | … | … |
| %% | | | | | | | | | | | | |
| Sector I | … | … | … | … | … | … | … | … | … | … | … | … |
| Sector II | … | … | … | … | … | … | … | … | … | … | … | … |
| Sector III | … | … | … | … | … | … | … | … | … | … | … | … |
| Sector IV | … | … | … | … | … | … | … | … | … | … | … | … |
| Sector V | … | … | … | … | … | … | … | … | … | … | … | … |
| **Norwai** | | | | | | | | | | | | |
| million euro | | | | | | | | | | | | |
| Sector I | … | 55,536.5 | 60,020.0 | 77,417.7 | 87,198.0 | 76,167.7 | 65,075.4 | 57,448.0 | 44,626.3 | 55,096.1 | 68,576.6 | 50,475.3 |
| Sector II | … | 19,172.0 | 22,251.2 | 23,150.3 | 25,102.4 | 24,759.0 | 24,599.3 | 22,986.0 | 21,075.4 | 21,429.9 | 20,700.4 | 21,118.3 |
| Sector III | … | 39,204.6 | 44,856.2 | 45,421.4 | 49,929. | 52,179. | 49,473.8 | 48,133.4 | 47,235.8 | 47,215.7 | 46,861.3 | 47,253.7 |
| Sector IV | … | 16,137.1 | 18,373.5 | 20,710.1 | 23,457. | 23,283.8 | 22,707.9 | 21,447.5 | 21,618.2 | 22,890. | 23,024.7 | 23,786.5 |
| Sector V | … | … | … | … | … | … | … | … | … | … | … | … |
| %% | | | | | | | | | | | | |
| Sector I | … | 30.35 | 29.03 | 33.23 | 33.41 | 30.04 | 27.52 | 26.19 | 22.20 | 25.58 | 30.03 | 23.22 |
| Sector II | … | 10.48 | 10.76 | 9.94 | 9.62 | 9.76 | 10.40 | 10.48 | 10.48 | 9.95 | 9.07 | 9.72 |
| Sector III | … | 21.42 | 21.70 | 19.50 | 19.13 | 20.58 | 20.92 | 21.94 | 23.50 | 21.92 | 20.52 | 21.74 |
| Sector IV | … | 8.82 | 8.89 | 8.89 | 8.99 | 9.18 | 9.60 | 9.78 | 10.75 | 10.63 | 10.08 | 10.94 |
| Sector V | … | 28.94 | 29.63 | 28.44 | 28.85 | 30.44 | 31.55 | 31.61 | 33.06 | 31.92 | 30.29 | 34.38 |

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
