# Peer review of "Sectoral Transformation of the Economic System during Crisis and Stable Growth Periods (A Case Study of the European Countries)"

_economies, doi:10.3390/economies10060148_

Round 1
Reviewer 1 Report
I do not think subparts in section 1 are needed. This is especially due to subsection 1.1 that is too short.
The section called Methodology should be renamed to Data and Methodology. Besides that I think the authors have to present in this section which are the countries used in the analysis as well. Actually, this is confusing. The tables have a total of 25 countries while Figure 1 9 countries, Table 4 and 5 present 7 countries, Figure 2 and 3 present 15 countries. There is nor clear and transparent presentation of the empirical results. The authors should present all countries in all figures and tables (even in an appendix) in order to give us a clear picture of all their results.
Why is the Gatev coefficient used? Are there any references which do so in the relevant literature. There must be presented some arguments on why the methodology employed is the one that is presented.
Why does the analysis stops on 2019. The pandemic could offer really important arguments for the scope of the paper and I think it is critical to use more recent data.
Figure 1 does not present the time in the horizontal axis. Moreover the authors in the title talk about the dynamics of oil production, while in the text about the dynamics in the share of the mining and quarrying industry!!!
The paper has a potential, but the authors have not been as focused as they should. I recommend major changes in order to give a second chance.
Author Response
Reviewer 1
I do not think subparts in section 1 are needed. This is especially due to subsection 1.1 that is too short.
We agree with the remark. Subsection numbers have been removed. Literature Review Section has been created in the manuscript.
The section called Methodology should be renamed to Data and Methodology. Besides that I think the authors have to present in this section which are the countries used in the analysis as well. Actually, this is confusing. The tables have a total of 25 countries while Figure 1 9 countries, Table 4 and 5 present 7 countries, Figure 2 and 3 present 15 countries. There is no clear and transparent presentation of the empirical results. The authors should present all countries in all figures and tables (even in an appendix) in order to give us a clear picture of all their results.
The Section has been renamed.
The number of analyzed countries is from 25 to 27 depending on the sectors; their initial number depends on the availability of data officially published by the statistical office of the European Union (Eurostat). In order to focus on the most important characteristics of economic dynamics, and not to overload the text with secondary information, Tables 4 and 5 and Figures 1, 2, 3 reflect the economic dynamics of those countries in which it manifests itself most significantly, reliably and clearly. At the same time, we agree with the reviewer's remark and provide data for all countries with analyzed dynamics of indicators in the Appendices:
Appendix 1: Dynamics of mining and quarrying production share in the structure of gross value added (GVA) at factor cost for 2008–2019.
Appendix 2:
2.1. Correlation between the volume dynamics of mining and manufacturing industries;
2.2. Correlation between the volume dynamics of industry and services in the total mining.
Appendix 3: Dynamics of sector indicators specific for the production of goods and services for 2008–2019.
Why is the Gatev coefficient used? Are there any references which do so in the relevant literature. There must be presented some arguments on why the methodology employed is the one that is presented.
In our opinion, this coefficient is optimal and illustrative for this type of calculation and is devoid of some shortcomings characteristic of other indicators. In particular, in contrast to the linear and quadratic coefficient of absolute structural shifts, the values of the Gatev coefficient vary from 0 to 1, and approaching 1 indicates notable differences in structural shifts. An alternative to the Gatev coefficient is the Salai coefficient, but its value alters greatly with the changes in values of the elements the totality is divided into. An application of various coefficients in calculations, and their advantages and disadvantages are clearly shown in regional studies of sectoral structures (Trifonov and Veselova 2015).
Why does the analysis stops on 2019. The pandemic could offer really important arguments for the scope of the paper and I think it is critical to use more recent data.
Sure, the pandemic can provide relevant arguments to identify new patterns and more convincing evidence (or refutation) of the former ones. However, at present it is not feasible to reliably use data on the socio-economic dynamics of the pandemics, because, firstly, the pandemic period has not yet completely ended, and secondly, unfortunately, Eurostat has not yet published statistical data for 2020-2022. Lacking these data, it is impossible to make calculations in one scale to form representative conclusions. In addition, the COVID-19 crisis was difficult to predict, unlike the expected 2008 crisis. Accordingly, the 2020-2023 pandemic-related crisis has its own features and unique manifestations. This issue will be studied by us in the future.
Figure 1 does not present the time in the horizontal axis.
It has been corrected.
Moreover the authors in the title talk about the dynamics of oil production, while in the text about the dynamics in the share of the mining and quarrying industry!!!
The error has been fixed.
The paper has a potential, but the authors have not been as focused as they should. I recommend major changes in order to give a second
Thank you for your comments and suggestions to improve our paper.

Reviewer 2 Report
My comments on the paper - Sectoral Transformation of the Economic System during Crisis and Stable Growth Periods (a Case Study of the European Countries) - are as follows.
The paper presents an interesting analysis and we consider that the research is of interest. However, as it stands, the paper needs revisions.
The introduction provides the necessary background information but the purpose of the paper should be indicated more explicitly.
The author does not mention the added value of the paper. In the introduction it must be stated the added value that the paper brings to the existing academic literature. What is the main contribution of this study?.
I suggest the author to create the Literature Review section.
The research methodology used by the author is adequate for the approached subject.
The interpretation of the results could be improved, in particular by explaining the causes (section 3).
What is the message of the conclusions?.
The conclusions do not underline the limits of the research. We consider that the author can show the limitations of the analysis carried out in his paper.
Author Response
Reviewer 2
My comments on the paper - Sectoral Transformation of the Economic System during Crisis and Stable Growth Periods (a Case Study of the European Countries) - are as follows.
The paper presents an interesting analysis and we consider that the research is of interest. However, as it stands, the paper needs revisions.
The introduction provides the necessary background information but the purpose of the paper should be indicated more explicitly.
The purpose of the work has been indicated more explicitly.
The purpose of the paper is to reveal and validate characteristic features of transformation regarding economic sectoral structure during the crisis of socio-economic system and the period of its systematic development in order to determine the most stable industry sectors.
The author does not mention the added value of the paper. In the introduction it must be stated the added value that the paper brings to the existing academic literature. What is the main contribution of this study?.
The added value of the paper has been stated in the Introduction.
Specification of regular transformations of the sectoral structure and identification of the most stable industry sectors under varying factors of economic dynamics (crisis and stable growth periods) are considered as the added value of the paper.
I suggest the author to create the Literature Review section.
Literature Review Section has been created in the manuscript.
The research methodology used by the author is adequate for the approached subject.
The interpretation of the results could be improved, in particular by explaining the causes (section 3).
An interpretation of the results for each stage of the analysis has been improved (Section 3). This is highlighted in the revised text of the manuscript.
What is the message of the conclusions?.
The conclusions do not underline the limits of the research. We consider that the author can show the limitations of the analysis carried out in his paper.
In our conclusions, we have tried to specify characteristic relationship between various sectors of production and services, their reaction to crisis manifestations and the specifics of behavior in the course of stable growth.
The limitations of the analysis are shown via the planning of future studies of economic dynamics, such as studying the specifics of using the principles of the fourth industrial revolution, and the features of the pandemic-related crisis.
However, we considered it appropriate to specify the limitations of our study and added the following paragraph:
«However, there are some limitations of our research. Its outcomes cannot be unambiguously applied either on a global scale or to any state. Within the framework of our research, we have shown that neither every European country is subject to the specific characteristics of the studied economic dynamics. It seems that even greater differences in economic behavior will be revealed in the analysis of transformations in similar periods in less developed and backward countries».
Thank you for your comments and suggestions to improve our paper.

Reviewer 3 Report
Thank you for review possibility of paper titled Sectoral Transformation of the Economic System during Crisis and Stable Growth Periods (a Case Study of the European Countries). It's interesting study, and I have some minor suggestions that can improve the paper:
It's hard to say what is the goal of the paper. It should be stated in Abstract and in the Introduction. Also the novelty of the study should be shown.
There are also other sectoral models like GICS used in sectoral analysis eg. DOI: 10.1016/j.cities.2011.09.005, DOI: 10.1016/j.geoforum.2013.09.015
There some studies about economic resilience of sectors and impact of economic crisis to sectoral development that should be added to the paper (eg. DOI: 10.3986/AGS.7416, DOI: 10.1007/s11769-017-0850-5)
good luck with your paper!
Author Response
Reviewer 3
Thank you for review possibility of paper titled Sectoral Transformation of the Economic System during Crisis and Stable Growth Periods (a Case Study of the European Countries). It's interesting study, and I have some minor suggestions that can improve the paper:
It's hard to say what is the goal of the paper. It should be stated in Abstract and in the Introduction. Also the novelty of the study should be shown.
The purpose of the work has been indicated more explicitly, and the added value of the paper has been stated.
The purpose of the paper is to reveal and validate characteristic features of transformation regarding economic sectoral structure during the crisis of socio-economic system and the period of its systematic development in order to determine the most stable industry sectors.
Specification of regular transformations of the sectoral structure and identification of the most stable industry sectors under varying factors of economic dynamics (crisis and stable growth periods) are considered as the added value of the paper.
There are also other sectoral models like GICS used in sectoral analysis eg. DOI: 10.1016/j.cities.2011.09.005, DOI: 10.1016/j.geoforum.2013.09.015
The recommended publications have been considered and included in the References; the GICS model has been studied and used in sectoral analysis in the manuscript.
There some studies about economic resilience of sectors and impact of economic crisis to sectoral development that should be added to the paper (eg. DOI: 10.3986/AGS.7416, DOI: 10.1007/s11769-017-0850-5)
The recommended publications have been considered and included in the References; the following paragraph has been added:
«Such periods are characterized by major changes in the nature of the socio-economic dynamics of states in comparison with the dynamics of coordinated development, and diversification of the behavior of economic entities. Said phenomena are viewed while analyzing territorial entities during major crises in dominant sectors in the total industry structure (Raźniak et al. 2017), and searching for prerequisites for the sustainability of cities and key industries in a crisis situation (Raźniak et al. 2020)».
good luck with your paper!
Thank you for your comments and suggestions to improve our paper.

Round 2
Reviewer 1 Report
I think the authors have improved the paper taking into account the comments.
Reviewer 2 Report
The author took into account the suggestions to the improved version of paper.